# Reinforcement Learning from Imperfect Corrective Actions and Proxy Rewards

**Zhaohui Jiang, Xuening Feng & Yifei Zhu**
University of Michigan-Shanghai Jiao Tong University Joint Institute, Shanghai Jiao Tong University
{jiangzhaohui,cindyfeng2019,yifei.zhu}@sjtu.edu.cn

**Paul Weng**
Digital Innovation Research Center, Duke Kunshan University
paul.weng@dukekunshan.edu.cn

**Yan Song, Tianze Zhou, Yujing Hu, Tangjie Lv & Changjie Fan**
NetEase Fuxi AI Lab
{songyan,zhoutianze,huyujing,hzlvtangjie,fanchangjie}@corp.netease.com

## Abstract

In practice, reinforcement learning (RL) agents are often trained with a possibly imperfect proxy reward function, which may lead to a human-agent alignment issue (i.e., the learned policy either converges to non-optimal performance with low cumulative rewards, or achieves high cumulative rewards but in undesired manner). To tackle this issue, we consider a framework where a human labeler can provide additional feedback in the form of corrective actions, which expresses the labeler's action preferences although this feedback may possibly be imperfect as well. In this setting, to obtain a better-aligned policy guided by both learning signals, we propose a novel value-based deep RL algorithm called **I**terative learning from **Co**rrective actions and **Pro**xy rewards (ICoPro)[1], which cycles through three phases: (1) Solicit sparse corrective actions from a human labeler on the agent's demonstrated trajectories; (2) Incorporate these corrective actions into the Q-function using a margin loss to enforce adherence to labeler's preferences; (3) Train the agent with standard RL losses regularized with a margin loss to learn from proxy rewards and propagate the Q-values learned from human feedback. Moreover, another novel design in our approach is to integrate pseudo-labels from the target Q-network to reduce human labor and further stabilize training. We experimentally validate our proposition on a variety of tasks (Atari games and autonomous driving on highway). On the one hand, using proxy rewards with different levels of imperfection, our method can better align with human preferences and is more sample-efficient than baseline methods. On the other hand, facing corrective actions with different types of imperfection, our method can overcome the non-optimality of this feedback thanks to the guidance from proxy rewards.

## 1 Introduction

While reinforcement learning (RL) has proved its effectiveness in numerous application domains (Mnih et al., 2015; Silver et al., 2017; Levine et al., 2016), its impressive achievements are only possible if a high-quality reward signal for the RL agent to learn from is available. In practice, correctly defining such reward signal is often very difficult (e.g., in autonomous driving). If rewards are misspecified, the RL agent would generally learn behaviors that are unexpected (Amodei et al., 2016b) and unwanted (Clark & Amodei, 2016; Russell & Norvig, 2016) by the system designer, notably due to overoptimization (Gao et al., 2023) or specification gaming (Randlov & Alstrom, 1998). This important issue has been well-recognized in the research community (Amodei et al., 2016a) and is an active research direction (Ouyang et al., 2022; Skalse et al., 2022).

---

[1]We have open-sourced its implementation: https://github.com/JiangZhaoh/ICoPro.

Various solutions have been proposed to avoid having to define a reward function, for instance, behavior cloning (Pomerleau, 1989; Bain & Sammut, 1995), inverse reinforcement learning (Ng & Russell, 2000; Russell, 1998), (inverse) reward design (Hadfield-Menell et al., 2017), or reinforcement learning from human feedback (Busa-Fekete et al., 2014; Christiano et al., 2017). However, these approaches could be impractical and inefficient since they may require a perfect demonstrator, a reliable expert to provide correct labels, or assume that the agent only learns from human feedback, which would subsequently require too many (often hard-to-answer) queries for a human to consider.

As an alternative intermediate approach, we propose the following framework in which the RL agent has access to two sources of signals to learn from: proxy rewards and corrective actions. A *proxy reward* function is an imperfect reward function, that approximately specifies the task to be learned. A *corrective action* is provided by a (possibly unreliable) human labeler when s/he is queried by the RL agent: a trajectory segment is shown to the labeler, who can then choose to correct an action performed by the agent by demonstrating another supposedly-good action.

This framework is practical and easily implementable. Regarding proxy rewards, they are generally easy for system designers to provide. For instance, they can (1) learn proxy rewards from (possibly imperfect) demonstration, (2) manually specify them to roughly express their intention (e.g., supposedly-good actions are rewarded and expected bad actions are penalized), or (3) only define sparse rewards (e.g., positive reward for reaching a destination and penalty for a crash). Regarding corrective actions, in contrast to typical demonstrations of whole trajectories, this feedback is usually much easier for the labeler to provide, since humans may not be able to complete a task themselves but can readily offer action preference on some states.

While only learning with one of those two sources of signals has its own limitations, by proposing our framework, we argue (and experimentally demonstrate) that learning simultaneously from both of them, even though both may be imperfect, can be highly beneficial. More specifically, on the one hand, solely learning from proxy rewards would either lead to very slow learning (e.g., when proxy rewards are well-aligned with ground-truth rewards but are very sparse) or yield a policy whose performance is not acceptable to the system designer (e.g., when proxy rewards are dense, but misspecified). On the other hand, solely learning from corrective actions would require too many queries to the human labeler and may lead to suboptimal learned behaviors since the human may be suboptimal. In contrast, our key insight is that the two sources of signals can complement each other. Since they are generally imperfect in different state-space regions, bad decisions learned from proxy rewards can be corrected by the human labeler, while the effects of suboptimal corrective actions may be weakened by proxy rewards. Therefore, learning simultaneously from the two imperfect signals can achieve better behaviors more aligned to the system designer's goals than using any one of the two alone and can be more sample-efficient (in terms of both environmental transitions and human queries).

As a proof of concept, we design a novel value-based RL algorithm (see Figure 1), called Iterative learning from Corrective action and Proxy reward (ICoPro). ICoPro alternates between three steps. In the first step (`datacollection`-phase), the RL agent interacts with the environment to collect transition data, and then the labeler provides corrective actions on them. In the second step (`finetune`-phase), the RL agent learns to select actions according to this feedback via a margin loss, which can be interpreted as an imitation learning (IL) loss. In the third step (`propagation`-phase), the RL agent is trained to maximize the expected cumulative proxy rewards while further enforcing a margin loss. This latter loss is expressed on both observed labels (i.e., corrective actions given by the human labeler) and pseudo-labels generated by a trained model (target Q-network). Imitating such pseudo-labels can be interpreted in two ways: either to reduce the number of queries or to stay close to the previously-learned policy, which can stabilize training like in trust region methods (Schulman et al., 2015; 2017; Asadi et al., 2022). By combining imitation learning and reinforcement learning in this third phase, the agent can learn from both proxy rewards and enforce the temporal consistency of the values learned from imitation learning.

The contributions of this paper can be summarized as follows: (1) We propose a practical human-in-the-loop reinforcement learning framework (Section 3) where the RL agent can learn from proxy rewards and (possibly suboptimal) corrective actions. (2) To train the agent from these two imperfect sources, we design an efficient and robust algorithm (Section 4) in which we demonstrate a simple but useful technique to reduce human feedback. (3) We experimentally validate our proof of concept (Section 5) by conducting an extensive number of experiments on various domains (e.g., image-

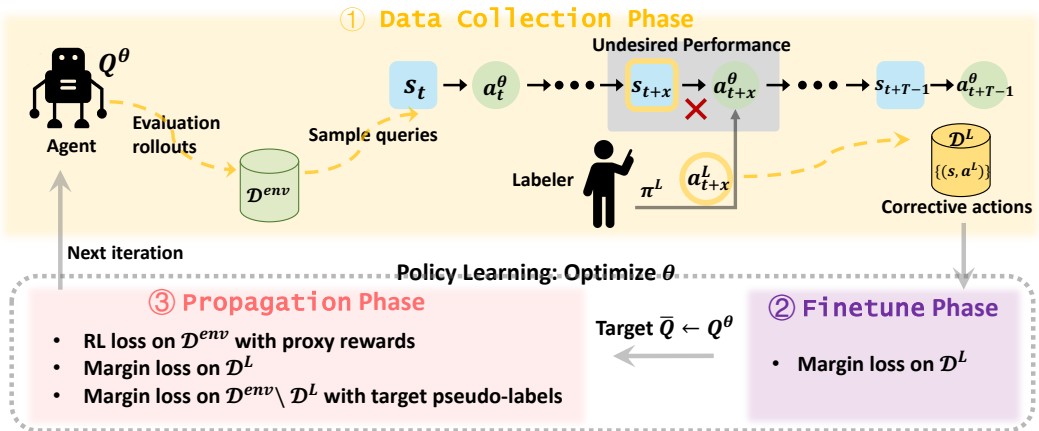

Figure 1: ICoPro is an iterative method with three phases in each iteration. It starts with the `data collection`-phase to collect agent's rollouts. Segments are then sampled from these rollouts and used as queries for the labeler to provide several corrective actions. Following this are two separate phases for policy updating, the `finetune`- and `propagation`-phase. Then the updated policy is utilized in the `data collection`-phase of the next iteration.

based Atari games and state-based highway driving with sparse and dense proxy rewards) under different conditions (e.g., simulated or real human labeler).

## 2 RELATED WORK

Given the difficulties in defining ground-truth rewards, a growing number of studies attempt to incorporate various types of human feedback to train agents, for example, demonstrations (Pomerleau, 1989; Hester et al., 2018; Reddy et al., 2019), preferences (Busa-Fekete et al., 2014; Christiano et al., 2017; Brown et al., 2020; Lee et al., 2021a), scalar evaluation (Knox, 2012; Knox & Stone, 2012; Saunders et al., 2018), action advising (Maclin & Shavlik, 1996; Da Silva et al., 2020; Ilhan et al., 2021a), interventions (Peng et al., 2023; Luo et al., 2024), example-states (Eysenbach et al., 2021), or combination of multiple feedback types (Ibarz et al., 2018; Jeon et al., 2020; Yuan et al., 2024; Dong et al., 2024). Among them, demonstrations, action advising, and interventions, which we discuss further below, are the most related to corrective actions. We roughly organize these works in three clusters. Preference signals can be provided in an *offline* or *online setting*. In addition, we discuss related work that considers *combining reward and other preference signals*.

**Offline setting** One of the earliest approaches to circumvent the need of defining a reward function is imitation learning (aka learning from demonstration), where a batch of demonstration is assumed to be available. Two main approaches have been developed: Behavior Cloning (BC) and Inverse Reinforcement Learning (IRL). BC (Pomerleau, 1989; Bain & Sammut, 1995) requires a significant amount of data to achieve the desired performance, while IRL (Ng & Russell, 2000; Russell, 1998; Brown et al., 2020) is computationally expensive as it derives the policy from the rewards obtained from demonstrations, although some work (Ho & Ermon, 2016; Reddy et al., 2019) aims to overcome these limitations. The major drawback in methods designed in the offline setting stems from training policies on a fixed limited dataset, which may yield a policy that can perform badly when deployed due to the train-test distribution mismatch. A typical approach to overcome this issue is to resort to online policy training, as we do in our method.

**Online setting** In contrast to the offline setting, here the agent learns with a human in the loop: either a human actively oversees the agent's training and can intervene, or the agent can query a human to obtain feedback to learn from. The former case (Spencer et al., 2020; Peng et al., 2023; Luo et al., 2024) requires active participation and continuous attention from humans, while the latter may require too many queries, which can be too labor intensive for one or even multiple humans to provide. Examples of the latter approach include methods in the DAgger family (Ross et al.,

2011; Sun et al., 2017; Kelly et al., 2019) (with optimal action labeling), and reinforcement learning from human feedback (RLHF) methods (Busa-Fekete et al., 2014; Christiano et al., 2017; Lee et al., 2021a) (with pairwise comparisons).

**Combining reward and other signals**   Since learning from one signal may be insufficient and/or inefficient, multiple works (Hester et al., 2018; Ibarz et al., 2018) consider learning from demonstration and rewards. In such work, demonstration is often provided offline and used for initializing an online training phase, which can be standard RL training (Hester et al., 2018) or use other types of feedback such as in RLHF (Ibarz et al., 2018), which can notably help for better human-agent alignment. In particular, demonstrations can be used to initialize a policy (Ibarz et al., 2018), define a reward signal (Reddy et al., 2019), or initialize a replay buffer for future RL training (Hester et al., 2018). In addition, they can also be queried online (Chen et al., 2020; Peng et al., 2023; Luo et al., 2024). Most works assume a well-specified (ground-truth) reward function, then focus on balancing the IL and RL objectives when human demonstrations are also usually assumed to be optimal (Shenfeld et al., 2023; Liu et al., 2024), although some propositions have been made to deal with suboptimal demonstrations (Nair et al., 2018).

In contrast to previous work, we do not assume the availability of a ground-truth reward function but only an approximation of it (i.e., a proxy reward), which is easier to specify. Since learning from this imperfect signal is insufficient, we complement it with corrective actions. Providing corrective actions is less cognitively-demanding for the human than generating whole demonstrations, especially if the agent and the human do not operate in the same sensori-motor space. Compared to action labeling (Ross et al., 2011; Kelly et al., 2019) or intervention (Peng et al., 2023; Luo et al., 2024), which have to happen on all unsatisfactory cases, our corrective action setting is less costly since the human chooses on which states s/he wants to give the feedback. Finally, compared to most previous work, we do not assume that the human is optimal. The goal of our paper is to demonstrate as a proof of concept that the agent can learn more efficiently to reach a better performance from two imperfect sources of signals than with any of the two alone.

## 3   PROBLEM FORMULATION

We consider a reinforcement learning (RL) problem (Sutton & Barto, 2018) where an RL agent repeatedly interacts with an environment: Given the state space $\mathcal{S}$ and action space $\mathcal{A}$, at a time step $t$, after observing an (observation) state $s_t \in \mathcal{S}$, it performs an action $a_t \in \mathcal{A}$, which yields an immediate reward $r_t \in \mathbb{R}$ and brings the agent to a new state $s_{t+1} \sim P(\cdot|s_t, a_t)$ where $P$ is the environmental transition function. The goal of the RL agent is to learn, from interaction tuples $(s_t, a_t, r_t, s_{t+1})$, a policy $\pi^* : \mathcal{S} \to \mathcal{A}$ that maximizes the expected discounted sum of rewards. Recall that a policy can be represented as the greedy policy with respect to a $Q$-function, where $Q : \mathcal{S} \times \mathcal{A} \to \mathbb{R}$ and $Q(s, a) = \mathbb{E}\left[\sum_{t=0}^{\infty} \gamma^t r_t\right]$ with discount factor $\gamma$, then $\pi(s) = \arg\max_a Q(s, a)$.

In contrast to standard RL, we do not assume that a *ground-truth* reward function (i.e., providing a perfect description of the task that the RL agent needs to learn) is available. Instead, the agent only receives an approximation of it, called *proxy reward* function and denoted $\widetilde{r}$. Indeed, while a precise reward function capturing all aspects of a desired behavior is hard to define for a system designer providing a proxy reward function to roughly express the system designer's objectives is much easier to achieve (Reddy et al., 2019; Jeon et al., 2020; Luo et al., 2024). However, since the proxy rewards are approximate, purely relying on learning from them may never achieve the desired performance. Therefore, we assume that the RL agent can also query a human labeler to obtain additional feedback under the form of *corrective actions* (see Figure 1). Formally, a *query* corresponds to a $T$-step segment (i.e., sequence of state-action tuples) $q = (s_t, a_t, \ldots, s_{t+T-1}, a_{t+T-1})$, which can for instance be sampled from trajectories generated from the interaction of the RL agent with its environment. Given such query, the human labeler may select one state $s_{t'}$ (such that $t \le t' \le t + T - 1$) where s/he provides a corrective action $a_{t'}^L = \pi^L(s_{t'})$, where $\pi^L$ is the labeler's policy. The corrective action is supposedly a better choice than the actually-performed one $a_{t'}$. Moreover, the labeler may choose to provide no feedback if all the actions in the segment are good.

Note that we do not assume that the labeler is necessarily optimal. This means that it is not possible to solely rely on the corrective actions (without proxy rewards) to learn a policy to achieve the desired performance, which would furthermore be impractical in terms of label collection cost for

the human labeler. Instead, by combining the two imperfect sources of learning signals, the RL agent could possibly learn a policy with a better performance and in a more sample-efficient way than using proxy rewards or corrective actions alone. Intuitively, the imperfections of the two signals probably do not lie in the same state-action space regions and therefore the two signals could correct each other. We confirm this intuition in our experiments (see Sections 5.2 and 5.3).

## 4 METHOD

As a first work demonstrating the benefit of learning from the two imperfect sources of feedback, we assume the action space $\mathcal{A}$ is finite for simplicity. Building on the Rainbow algorithm (Hessel et al., 2018), we design ICoPro, an iterative approach to learn $Q^\theta$ with desired policy $\pi^\theta(s) = \arg\max_{a \in \mathcal{A}} Q^\theta(s, \cdot)$. At each iteration $i$, ICoPro updates the current $Q^{\theta_i}$ to obtain the next $Q^{\theta_{i+1}}$ according to the three phases: `DataCollection`, `Finetune`, and `Propagation`, which we elaborate on next. Figure 1 outlines the proposed framework and Algorithm 1 is provided in Appendix B.1.

**Data Collection-phase.** Inspired by the popular RLHF setting (Christiano et al., 2017), we adopt the following protocol for collecting two types of data: interaction tuples and corrective actions. For the former, the agent acts in its environment according to the $\epsilon$-greedy policy with respect to $Q^{\theta_i}$, denoted $\mathcal{G}_\epsilon(Q^{\theta_i})$, to collect a set $\mathcal{D}_i^{env}$ of $H$ interaction tuples, $(s_t, a_t, \widetilde{r}_t, s_{t+1})$, which are also stored in the transition replay buffer $\mathcal{D}^{env}$. For the latter, the RL agent can regularly during RL training issue a batch of sampled queries from $\mathcal{D}_i^{env}$ to the labeler to obtain a set $\mathcal{D}_i^L$ of pairs of state and corrective action $(s, a^L)$, which are also stored in the feedback replay buffer $\mathcal{D}^L$. The collected data, $\mathcal{D}^{env}$ and $\mathcal{D}^L$, are used to train the RL agent in the next two phases. As a side remark, for simplicity, querying the labeler is currently implemented in our experiments in a synchronous way (i.e., the agent waits for the labeler's feedback), but this process could possibly also be asynchronous.

**Finetune-phase.** This phase is a pure supervised training phase to learn from all the labeler's corrective actions collected so far in $\mathcal{D}^L$. Formally, $Q^{\theta_i}$ is updated with the following margin loss $\mathcal{L}_L^{MG}$ (Kim et al., 2013; Piot et al., 2014; Hester et al., 2018; Ibarz et al., 2018):

$$\mathcal{L}^{MG}(\theta_i \mid \mathcal{D}^L) = \mathbb{E}_{(s, a^L) \in \mathcal{D}^L} \left[ \max_{a \in \mathcal{A}} \left[ Q^{\theta_i}(s, a) + l(a^L, a) \right] - Q^{\theta_i}(s, a^L) \right], \qquad (1)$$

where $l(a^L, a) = 0$ if $a = a^L$, and a non-negative *margin* value $C$ otherwise. This loss amounts to enforcing that the corrective actions' Q-values should not be smaller than those of any other actions. As common practice, $Q^{\theta_i}$ is updated via mini-batch stochastic gradient descent using Equation (1). Following classical methods (Christiano et al., 2017) in RLHF, the updates end when the actions predicted by the updated $Q^{\theta_i}$ reaches a pre-defined accuracy $\delta_{acc}$: $\mathbb{P}_{s \sim \mathcal{D}^L} \left[ a^L = \arg\max_a Q^{\theta_i}(s, a) \right] > \delta_{acc}$.

**Propagation-phase.** Since pure imitation learning requires a large number of human labels, we design the `propagation`-phase to include the training of the updated $Q^{\theta_i}$ with a combination of RL losses and margin loss (from actual labels in $\mathcal{D}^L$ but also pseudo-labels), which we explain next.

*- Training with RL Losses* allows not only learning from proxy rewards, but also propagating the effect of human labels to more states (learned in the previous phase). The initial target $\bar{Q}$ in this phase is the $Q^{\theta_i}$ obtained at the end of the `finetune`-phase. These RL losses are composed of two commonly used terms, $\mathcal{L}_1^{RL}$ and $\mathcal{L}_n^{RL}$, which are respectively defined as:

$$\mathcal{L}_1^{RL}(\theta_i \mid \mathcal{D}^{env}, \bar{Q}) = \mathbb{E}_{\mathcal{D}^{env}} \left[ \left( Q^{\theta_i}(s_t, a_t) - \left( \widetilde{r}_t + \gamma \max_{a' \in \mathcal{A}} \bar{Q}(s_{t+1}, a') \right) \right)^2 \right] \text{ and} \qquad (2)$$

$$\mathcal{L}_n^{RL}(\theta_i \mid \mathcal{D}^{env}, \bar{Q}) = \mathbb{E}_{\mathcal{D}^{env}} \left[ \left( Q^{\theta_i}(s_t, a_t) - \left( \sum_{k=0}^{n-1} \gamma^k \widetilde{r}_{t+k} + \gamma^n \max_{a' \in \mathcal{A}} \bar{Q}(s_{t+n}, a') \right) \right)^2 \right], \qquad (3)$$

where $\mathcal{D}^{env}$ is the transition replay buffer, and $\bar{Q}$ is the target network from a historical version of $Q^{\theta_i}$. Note that in the context of standard RL, Equation (3) is actually not theoretically-founded,

since the goal is to learn a greedy policy. However, in our context, it can help propagate faster the Q-values learned from the corrective actions.

**- *Training with margin loss using both actual and pseudo-labels*** allows to complement and correct the training with proxy rewards. Since the corrective actions in $\mathcal{D}^L$ are not executed in $\mathcal{D}^{env}$, training with them using loss $\mathcal{L}^{MG}$ prevents the agent from forgetting about the actual labels. In addition, training with pseudo-labels can reduce the cost of collecting human labels by leveraging the large number of unlabeled states in $\mathcal{D}^{env}$. Pseudo-labels can be generated, using predicted greedy actions from the target $\bar{Q}$, only on unlabeled states, since otherwise, it is better to enforce the actual label than a pseudo-label. Formally, the margin loss using pseudo-labels can be expressed as follows:

$$\mathcal{L}_{TGT}^{MG}(\theta_i \mid \mathcal{D}^{TGT}) = \mathbb{E}_{(s,a^{TGT})\in\mathcal{D}^{TGT}}\left[\max_{a\in\mathcal{A}}\left[Q^{\theta_i}(s,a) + l(a^{TGT},a)\right] - Q^{\theta_i}(s,a^{TGT})\right], \quad (4)$$

where $\mathcal{D}^{TGT} = \{(s,a^{TGT}) \mid s \in \mathcal{D}^{env}\backslash\mathcal{D}^L, a^{TGT} = \arg\max_{a\in\mathcal{A}}\bar{Q}(s,\cdot)\}$ and by abuse of notations, $\mathcal{D}^{env}\backslash\mathcal{D}^L$ denotes the states that have not received any actual corrective actions $a^E$. Intuitively, as the training process goes on, we expect that the quality of the pseudo-labels improves, enhancing further the benefit of enforcing $\mathcal{L}_{TGT}^{MG}$. As a side note, $\mathcal{L}_{TGT}^{MG}$ can also be understood as regularizing the online network's parameters to stay close to the target network, which has been shown to be beneficial in terms of learning efficiency (Asadi et al., 2022). In contrast to this previous work, we regularize via the margin loss in the policy space instead of the parameter space.

To sum up, the total loss in this phase is the sum of the RL losses and the margin losses[2]:

$$\mathcal{L}^{Prop} = \mathcal{L}_1^{RL}(\theta_i \mid \mathcal{D}^{env}, \bar{Q}) + \mathcal{L}_n^{RL}(\theta_i \mid \mathcal{D}^{env}, \bar{Q}) + \mathcal{L}_{L+TGT}^{MG}(\theta_i \mid \mathcal{D}^{env}, \mathcal{D}^L),$$

$$\text{where } \mathcal{L}_{L+TGT}^{MG}(\theta_i \mid \mathcal{D}^{env}, \mathcal{D}^L) = (1-\bar{w})\mathcal{L}_L^{MG}(\theta_i \mid \mathcal{D}^L) + \bar{w} \cdot \mathcal{L}_{TGT}^{MG}(\theta_i \mid \mathcal{D}^{env}), \quad (5)$$

and hyperparameter $\bar{w}$ controls how much weight is put on pseudo-labels. In our implementation, $\mathcal{L}_1^{RL}$, $\mathcal{L}_n^{RL}$ and $\mathcal{L}_{TGT}^{MG}$ works on a same minibatch from $\mathcal{D}^{env}$ in each gradient step, while $\mathcal{L}_L^{MG}$ works on another one of the same size from $\mathcal{D}^L$ to relieve the unbalanced issue of these two datasets.

## 4.1 OTHER IMPLEMENTATION DETAILS

**Updating schedule for $Q^\theta$ and $\bar{Q}$.** Since $a^{TGT}$ will change every time we update the target $\bar{Q}$, we update $\bar{Q}$ with low frequency to stabilize training. Concretely, there are $E$ epochs in one `propagation`-phase that each epoch optimize $\mathcal{L}^{Prop}$ through $\mathcal{D}^{env}$. At the end of each epoch, $\bar{Q}$ will be updated to the online parameter $Q^{\theta_i}$. If $E = 1$, pseudo-labels in `propagation`-phase are generated form the policy at the end of `finetune`-phase. If $E > 1$, pseudo-labels are changing after each epoch and can come from the (potentially) improved $Q^\theta$ after optimizing $\mathcal{L}^{Prop}$.

**Optimization for ICoPro's iterative scheme.** As for the optimizer, we use use a same optimizer through the two learning phases in ICoPro. Since ICoPro uses an iterative learning scheme, we follow the research from Asadi et al. (2024) that reset the 1st and 2nd order moment inside the Adam optimizer at the beginning of each learning phase. The `finetune`- and `propagation`-phase use the same learning rate $\alpha$, which remains unchanged throughout the training procedure.

## 5 EXPERIMENTS

We first explain our general experimental settings in Section 5.1. To compare ICoPro with baselines and analyze its design components on a large scale of experiments, we use simulated labelers to demonstrate and analyze ICoPro's performance in Sections 5.2 and 5.3. Then in Section 5.4, we further verify the alignment performance of ICoPro using real humans' action feedback.

## 5.1 GENERAL EXPERIMENTAL SETUPS

To ensure reasonable labeling effort, only at most 1% of the environmental transitions are labeled in our experiments. Hyper-parameter values are given in Appendix B.2 unless otherwise specified. Detailed evaluation configurations can be checked in Appendix D.1.

---

[2]In Equation (5) we give equal weight to the IL and RL losses, but unequal weights could be provided if we expect one source to be more suboptimal than the other.

**Baselines.** To compare with the performance of learning from only proxy rewards, we use Rainbow (Hessel et al., 2018) as a baseline. To compare with the performance of learning from only corrective actions using $\mathcal{L}_L^{MG}$ (Equation (1)), we evaluate behavior cloning (BC-L) on all collected labels $\mathcal{D}^L$ at the end of training of ICoPro as the one in offline style, and adapt HG-DAgger (Kelly et al., 2019) (HGDAgger-L) by removing the `propagation`-phase from ICoPro as the one in online style. As for baselines involving learning from both signals, since no previous work considers the exact same learning scheme as explained in Section 3, we adapt two state-of-the-art methods that were initially designed on normal RL methods into our iterative learning setting with the same labeling schedule and budget, DQfD (Hester et al., 2018) (DQfD-I) and PVP (Peng et al., 2023) (PVP-I), to compare the performance when learning from both corrective actions and proxy rewards. Basically, the adaption is canceling the `finetune`-phase in ICoPro and replacing their loss functions with ICoPro's $\mathcal{L}^{Prop}$ (Equation (5)) in the `propagation`-phase. DQfD-I can be seen as an ablation of ICoPro by removing $\mathcal{L}_{TGT}^{MG}$ from $\mathcal{L}^{Prop}$. PVP-I's loss replaces $\mathcal{L}_{L+TGT}^{MG}$ to $\mathcal{L}_L^{PVP} = \mathbb{E}_{(s,a,a^E)\sim\mathcal{D}^L}\left[|Q^\theta(s,a^E) - 1|^2 + |Q^\theta(s,a) + 1|^2\right]$, but using zero rewards in $\mathcal{L}_1^{RL}$ and $\mathcal{L}_n^{RL}$. We also test in PVP-I, the effects of using proxy rewards instead of zeros (PVP-IR). More details about the baselines and other ablations mentioned in Section 5.2 can be checked in Appendix C.1.

**Environments.** Our experiments involve both the state-based highway (Leurent, 2018) environment and imaged-based Atari (Aitchison et al., 2023) environments.

Highway is an environment that allows for flexible design of diverse proxy rewards and evaluation metrics, but is relatively simpler than Atari in terms of action dimension ($|\mathcal{A}| = 5$) and environmental complexity. The basic goal in this environment is to drive a car on a straight road with multiple lanes. The final performance of the trained vehicle can be measured with different metrics, e.g., the ratio of episodes with a crash (%Crash), the average total forward distance in a given time limit (Distance-Avg), the average speed (Speed-Avg), and the ratio of taking a change lane action (%LaneChange-Avg). We design representative proxy rewards in different levels of imperfection by associating events with different rewards as shown in Table 1.

Table 1: Engineered proxy rewards on highway. We normalize the rewards into the range $[-1, 1]$ in each step by min-max scaling if Normalization is not None, where the min (max) reward is the sum of negative (positive) event rewards.

| Configurations | | Proxy Rewards | | | | |
|---|---|---|---|---|---|---|
| | | PRExp | PR1 | PR2 | PR3 | PR4 |
| Rewards for Events | Change lane | 0.2 | 0 | 0.2 | 0 | 0 |
| | High speed | 1.5 | 2 | 0.8 | 0 | 0 |
| | Low speed | -0.5 | -1 | 0 | 0 | 0 |
| | Crash | -1.7 | -1 | -1 | -1 | -1 |
| Normalization | | [-1,1] | [-1,1] | [-1,1] | [-1,1] | None |
| Dense(D) or Sparse(S) | | D | D | – | D | S |

In contrast, Atari provides more complex environments with larger action dimensions and longer episode lengths, although with less flexibility in designing proxy rewards and diverse performance metrics. Since the dimension of action space can be seen as a measurement of hardness for one game, we first select 6 representative games with the full action dimension ($|\mathcal{A}| = 18$) across various categories suggested by Aitchison et al. (2023): (1) Combat: Battlezone and Seaquest, (2) Sports: Boxing, (3) Maze: Alien, (4) Action: Frostbite and Hero. We also include smaller-action-dimensional but classical games to complement our evaluation: MsPacman and Enduro ($|\mathcal{A}| = 9$), Pong ($|\mathcal{A}| = 6$), and Freeway ($|\mathcal{A}| = 3$). In this setting, cumulative episodic raw rewards from ALE are used to measure agent performance, with signed raw rewards serving as proxy rewards[3].

**Simulated labeler.** We use Q-values trained with Rainbow to simulate human feedback with $Q^{diff}(s, a, a^L) = Q^L(s, a^L) - Q^L(s, a)$, where $Q^L(s, a) = \mathbb{E}_{P,\pi^L}[\sum_{t=0}^{\infty} \gamma^t r^L(s, a)|s_0 = s, a_0 = a]$ is the labeler's Q-function, and $a^L \sim \mathcal{G}_\epsilon(Q^L(s, \cdot))$ is the action issued from the labeler, $a \sim \mathcal{G}_\epsilon(Q^\theta(s, \cdot))$ is the executed action from the training agent. In practice, we select states with top $N_{CF}$ largest $Q^{diff}$ to give corrective actions for a querying segment Luo et al. (2024) , where $N_{CF} = 1$ is our default configuration. Basically, simulated labelers are agent checkpoints trained with engineered rewards by Rainbow but do not necessarily converge to optimal performances. For highway, labelers are trained with PRExp mentioned in Table 1. For Atari, labelers are trained with the signed raw rewards. Concrete configurations to train labelers can be checked in Appendix D.4.

---

[3]Raw rewards $r_A$ in Atari can be seen as goal-conditioned rewards, but not all desired performances (e.g., achieve goals in the right way) are rewarded in $r_A$. Further discussion is available in Appendix D.3.

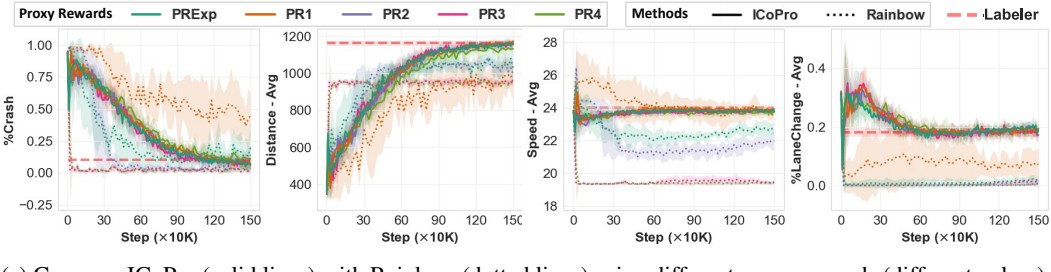

(a) Compare ICoPro (solid lines) with Rainbow (dotted lines) using different proxy rewards (different colors).

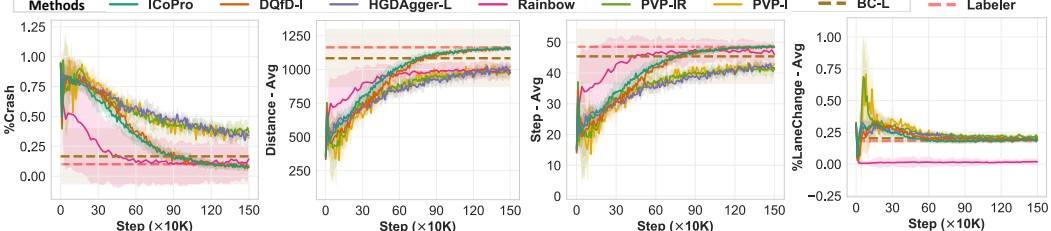

(b) Compare ICoPro with other baselines (different colors). Results are averaged over the set of proxy rewards.

Figure 2: Experiments on highway using the set of proxy rewards mentioned in Table 1. Each subplot compares the performance with respect to one representative performance metric. $|\mathcal{D}^L|$=1.5K.

Table 2: Experimental results on Atari with different size of $|\mathcal{D}^L|$. Red fonts denote the best results with the largest mean score. Bold black fonts emphasize results whose mean+std covers the best mean score. Yellow backgrounds highlight results that match the labelers' performance in the sense that their mean scores are in the range of the labelers' mean±std.

| $|\mathcal{D}^L|$ | Methods | Seaquest | Boxing | Battlezone | Frostbite | Alien | Hero | MsPacman |
|---|---|---|---|---|---|---|---|---|
| \ | Labeler | 1071.92±224.88 | 66.05±9.17 | 24485.19±4987.60 | 5647.63±1038.91 | 2908.97±908.32 | 26866.07±612.29 | 3539.54±1270.95 |
| | Rainbow | 707.04±98.96 | 1.99±0.71 | 17058.00±2310.60 | 2766.40±518.78 | 1319.83±457.06 | 19413.57±600.39 | 1773.98±323.19 |
| Large | HGDAgger-L | 790.10±73.98 | 24.95±2.22 | 16573.33±1262.82 | 3207.35±527.35 | 2067.45±262.04 | 13258.79±2128.13 | **3052.38**±370.50 |
| | PVP-I | 635.83±65.74 | 7.76±0.84 | 15183.03±1739.17 | 1535.28±528.88 | 596.56±99.45 | 8503.46±1205.13 | 1686.13±608.96 |
| | PVP-IR | 770.88±91.00 | 13.36±2.00 | 17378.18±2245.64 | 1664.62±414.02 | 798.55±137.99 | 7729.12±1282.99 | 920.25±216.76 |
| | BC-L | 758.40±41.31 | 21.39±1.21 | 16020.00±2081.92 | 3410.27±479.08 | 2360.40±422.57 | 14288.67±1673.48 | 2877.07±279.24 |
| | w/o $a^{TGT}$ | **1213.98**±102.54 | 48.19±7.90 | 18111.52±1409.68 | 3874.36±515.30 | 1835.06±267.85 | 17805.35±2471.35 | 2579.86±186.22 |
| | w/o Finetune | **1234.39**±113.44 | 51.83±26.67 | 19366.67±285.28 | **4713.07**±1211.91 | 2223.66±368.38 | 8398.37±2729.85 | 2633.59±154.34 |
| | DQfD-I | 1155.43±85.98 | 41.71±5.09 | 18444.85±1279.70 | 4176.27±659.14 | 2079.99±293.53 | 17814.10±1688.55 | 2734.25±176.22 |
| | ICoPro | 1274.76±109.53 | 61.52±2.47 | 20030.30±1620.36 | 4817.74±605.45 | 2360.27±301.52 | 23344.01±2353.67 | 3188.20±453.13 |
| Small | HGDAgger-L | 508.34±77.91 | 17.91±2.21 | 14682.22±1633.46 | 1561.04±457.61 | 1582.74±467.88 | 10617.77±1664.62 | **2342.51**±264.97 |
| | PVP-I | 439.96±68.62 | 1.09±2.05 | 9242.42±2056.15 | 509.45±237.64 | 402.53±78.44 | 6454.75±1193.92 | 891.84±251.87 |
| | PVP-IR | 733.73±116.70 | 14.33±3.98 | 16510.30±1684.40 | 1883.64±270.40 | 1136.07±301.34 | 8777.76±1410.76 | 1097.92±246.94 |
| | BC-L | 537.20±51.86 | 9.09±4.64 | 17273.33±2605.40 | 1348.80±236.28 | 1711.00±659.06 | 10252.73±1556.18 | **2278.00**±337.84 |
| | w/o $a^{TGT}$ | **1065.65**±164.01 | 40.56±10.22 | 16465.45±1434.87 | 2785.41±452.83 | 1269.06±288.76 | 15420.19±2108.08 | 1676.97±652.21 |
| | w/o Finetune | **1081.16**±131.98 | 19.35±16.53 | 21253.33±2694.61 | **3037.84**±1152.24 | 891.97±148.50 | 9759.02±3093.64 | 1830.23±559.33 |
| | DQfD-I | 1003.21±128.45 | 39.27±20.85 | 17003.64±1532.05 | 3146.25±418.00 | 1075.79±249.17 | 16525.18 ±1915.10 | 1952.64±427.78 |
| | ICoPro | 1167.19±103.02 | 61.40±23.91 | 19969.09±1555.69 | 3368.31±701.54 | 1736.52±513.91 | 17862.58±2181.02 | 2558.57±331.18 |

## 5.2 SAMPLE-EFFICIENT ALIGNMENT WITH IMPERFECT PROXY REWARDS

We use one scripted labeler for each environment to validate the alignment ability of ICoPro and compare it with baselines, using proxy rewards with different levels of imperfections. In the following of this part, we analyze ICoPro's performance based on results shown in Figure 2 and Table 2. More specifically, Figure 2 evaluates the performance with various performance metrics, while Table 2 evaluates the performance with small or large size of action label budget[4]. Due to space constraints, results for easier Atari tasks (i.e., Pong, Freeway, and Enduro) and corresponding plots for the results in Table 2, are put into Appendix A.

---

[4]Considering that the action budgets required to align depend on the complexity of the labeler's performance, the concrete values of $|\mathcal{D}^L|$ is not specified in the main text but can be checked in Table 7.

**Learning from a combination of two imperfect signals performs better than learning from $\widetilde{r}$ or corrective actions alone.** Compared to Rainbow, Figure 2a and Table 2 show that ICoPro with corrective actions can converge quickly and stably to the labeler's performance using proxy rewards with different levels of imperfection, but Rainbow's performance is unstable and can easily converge to non-optimal performance without the guidance from corrective actions. Compared to BC-L and HGDAgger-L, Figure 2b and Table 2 demonstrate that ICoPro's performance exceeds their performances substantially in most of the games, which confirms the effectiveness of integrating the proxy rewards with the RL losses to achieve better alignment than using corrective actions alone.

**ICoPro achieves better alignment performance than baselines.** As shown in Figure 2b and Table 2, using the same sample budget in terms of both environmental transitions and label budget, ICoPro achieves the best performance in terms of aligning with labelers' performance in almost all settings, or performs similarly with the best one otherwise. In Figure 2b, except for Rainbow, all methods perform similarly in terms of aligning the labeler's Speed-Avg and %LaneChange-Avg performance, while for the harder performance metrics %Crash and Distance-Avg, ICoPro and DQfD-I perform similarly, and both are significantly better than PVP-I, PVP-IR, and HGDAgger-L. In Table 2, with large $\mathcal{D}^L$, ICoPro does match the labelers' performances, except in the hardest Hero with full action dimensions and relatively long episode length. Specifically, compared with the two PVP methods, their performances are obviously worse than that of the methods using margin loss, since $\mathcal{L}_L^{PVP}$ sets a pre-defined bound for Q-values making it fails to adapt to various reward settings. In some games, incorporating proxy rewards into their framework (PVP-IR) performs even worse than not (PVP-I). However, margin loss is a suitable choice to incorporate the corrective actions with proxy rewards. As for DQfD-I, in simple environments we perform similarly, but in harder Atari environments ICoPro outperforms it and shows more notable performance gaps than in highway.

**Ablations: ICoPro's integral design leads to a stable and robust performance.** Considering that the two-phase learning and pseudo-labeling are the main novel designs in ICoPro, there are two additional settings to be examined besides DQfD-I and HGDAgger-L: (1) w/o `Finetune`, which removes the `Finetune`-phase but retains the pseudo-labels, and (2) w/o $a^{TGT}$, which removes pseudo-labels[5]. In the highway environment, the two ablations perform similarly to ICoPro, and we include them into Appendix A.2 instead of Figure 2b to make it clearer to check. However, in Table 2, we find that while the two ablation settings may perform well in individual environments when action feedback is abundant, they are not as robust as ICoPro across different environments, especially when feedback is limited and tasks are challenging. ICoPro consistently demonstrates superior performance in such scenarios, maintaining its lead even with fewer feedback instances.

### 5.3 Overcoming Non-optimality of Corrective Actions with Proxy Rewards

To simulate different types of imperfections inside corrective actions, the most natural and commonly adopted (Lee et al., 2021b; Luo et al., 2024) one is to replace part of a labeler's corrective actions with random actions (`DiffRand`). In our setting, it simulates labelers who know which state-action pairs to correct but provide noisy corrective actions. We evaluate it in three representative environments that cover various environmental complexities: highway, Boxing and Seaquest. As shown in Figures 3a and 3b. ICoPro performs robustly within a reasonable ratio of random actions (i.e., $< 25\%$), and can perform significantly better than the labelers with large randomness (i.e., $50\%$). For Atari, we further consider a harder type that uses different labelers with worse performances (`DiffLabler`) to simulate labelers who are suboptimal on both where and how to provide corrective actions. As shown in Figure 3c, ICoPro still demonstrate a strong capability to overcome the non-optimality inside such unreasonable action feedback in most of the cases. Performance of Highway using another labeler, and training plots with concrete budget sizes or proxy rewards, instead of the averaged ones in Figure 3 can be checked in Appendices A.1.3, A.2.2 and A.2.3.

### 5.4 User study

To verify that ICoPro can align with real human's preferences when the desired performance is hard to be obtained with engineered rewards, we use highway ($|\mathcal{A}| = 5$), Pong ($|\mathcal{A}| = 6$), and Seaquest ($|\mathcal{A}| = 18$) as three representative environments to evaluate *ICoPro with real human*

---

[5]Considering that removing margin loss on $\mathcal{D}^L$ from the `propagation`-phase is not reasonable, we do not show this set of ablations in the main text, but the results can be checked in Appendix C.3.1.

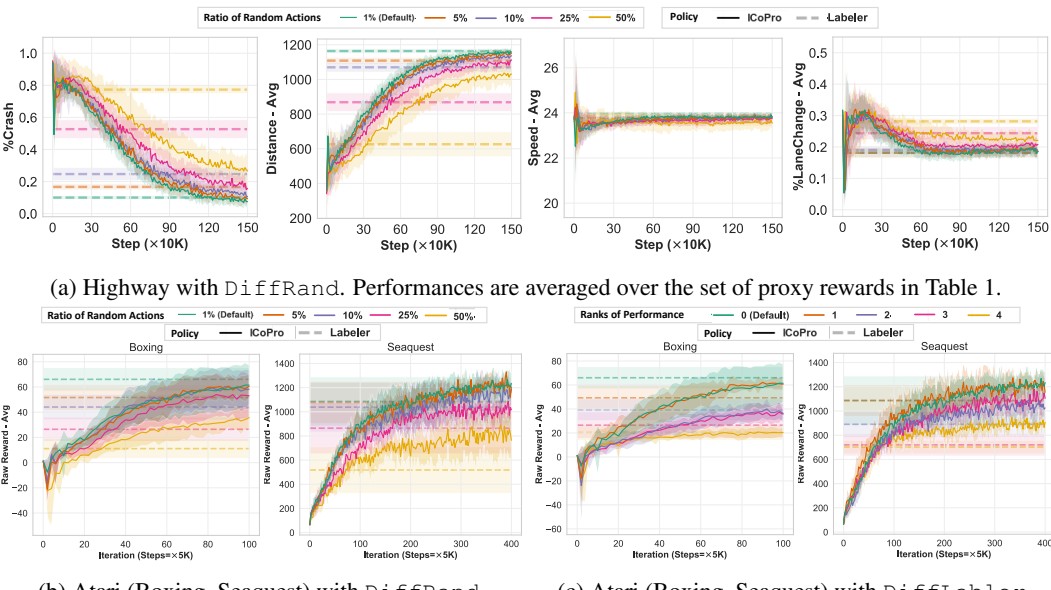

(a) Highway with `DiffRand`. Performances are averaged over the set of proxy rewards in Table 1.

(b) Atari (Boxing, Seaquest) with `DiffRand`.      (c) Atari (Boxing, Seaquest) with `DiffLabler`.

Figure 3: ICoPro facing different types of non-optimality of corrective actions (different colors). Performances are averaged over large/small $\mathcal{D}^L$ in Figures 3b and 3c.

(ICoPro-Human) with less than 500 action labels. For highway, we use PR4 as the proxy reward in ICoPro-Human since it does not introduce any extra performance shaping, and obtain a vehicle that knows how to take over cars with super fast average speed ([Speed-Avg,%Crash]=[29.02, 0.22]), which is a behavior that we could never obtain with Rainbow using PR4 alone. For Pong, ICoPro-Human obtains an agent that performs more human-like than the simulated optimal labelers (see Figure 12 in Appendix E.1 for details). For Seaquest, we use it to further verify the applicability of ICoPro-Human to environments with larger action dimensions. In this game, saving divers is a desired performance that has never been correctly rewarded in its proxy rewards. ICoPro-Human obtains an agent that rescues 1.5 times more divers than the scripted labeler (3.2 vs. 1.9, averaged over 50 evaluation episodes), even though the scripted labeler costs 46 times more timesteps (0.1M vs. 4.6M). Moreover, compared to ICoPro-Human, BC-L on those real human labels gets poor performance due to the limited data and non-optimality of human feedback (highway: [29.02, 0.22] vs. [27.90, 0.41]), Pong: 7.83 vs. -11.26, Seaquest: 3.2 vs. 2.1), which confirms again ICoPro's ability to learn from two imperfect signals in a sample-efficient way. User interfaces and instructions for each environment, examples of corrective actions from humans, as well as the training plots and evaluation videos of ICoPro-Human, are provided in Appendix E.

## 6 CONCLUSIONS AND LIMITATIONS

We present a human-in-the-loop framework where an RL agent can learn from two potentially unreliable signals: corrective actions provided by a labeler and a pre-defined proxy reward. The motivation for combining the two learning signals is threefold: (1) guide the agents' training process with corrective actions, (2) use proxy rewards to help reduce human labeling efforts, and (3) possibly compensate for the imperfection of one signal with the other. Our value-based method trains a policy in an iterative way, with two separate learning phases inside each iteration: the `finetune`-phase uses a margin loss to update the policy to better align with action feedback, then the `propagation`-phase incorporates RL losses as well as the margin loss expressed with pseudo-labels to generalize the improved values. Our experiments validate our algorithmic design.

While we demonstrated the feasibility and the benefit of learning from two imperfect signals, we plan as future work to provide a more theoretical analysis to reveal what assumptions about the misspecification of proxy rewards and/or the suboptimality of the corrective actions could guarantee their synergetic combination.

## 7 ACKNOWLEDGEMENT

This work has been supported in part by the National Key R&D Program of China (Grant No. 2024YFC3017100) and the program of the National Natural Science Foundation of China (No. 62176154).

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

# A MORE EXPERIMENTAL RESULTS AND PLOTS

## A.1 ATARI

### A.1.1 LEARNING CURVES FOR TABLE 2 INCLUDING PONG, FREEWAY, AND ENDURO

Figure 4 show plots comparing ICoPro with various baselines, and Figure 5 show ablation study for ICoPro. Lines represent the average episodic return in terms of the raw reward. Shadows represent the standard deviation. For Rainbow, the x-axis should read in terms of steps instead of iterations. For other methods, the x-axis can be read in terms of steps or iterations.

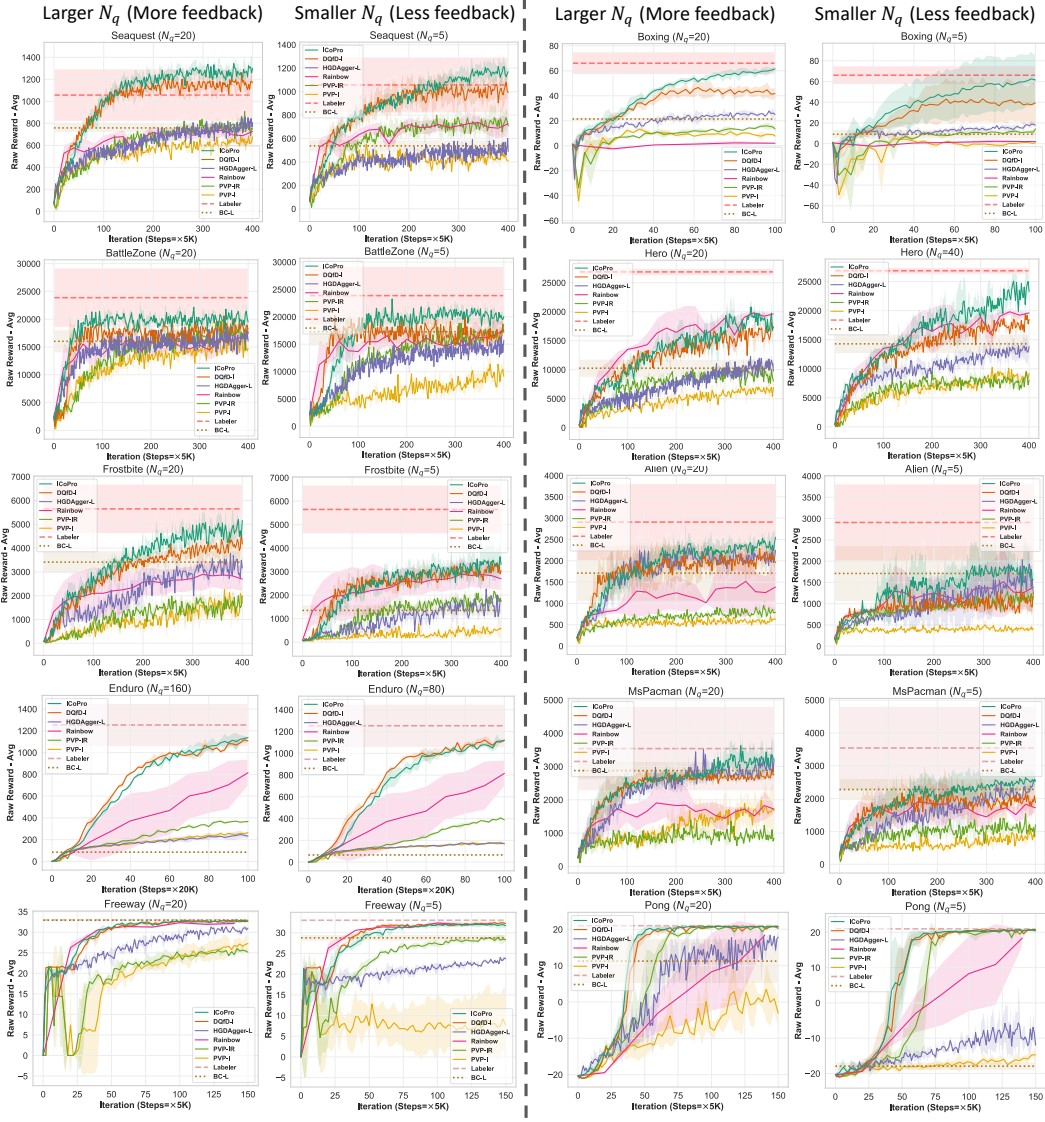

Figure 4: Compare baseline methods on Atari in terms of the averaged episode return measured with the raw reward $r_A$. The shadow indicates the standard deviation over 5 seeds. $N_q$ in titles refer to the number of queries per iteration, and the larger (resp. smaller) ones correspond to the large (resp. small) $\mathcal{D}^L$ in Table 2.

### A.1.2 COMPARE ICOPRO WITH MORE RELATED WORKS.

**Compare with Ilhan et al. (2021b).** In Table 3, we compare the performance of ICoPro with ActionImitation (Ilhan et al., 2021b), which is a method that also uses action advice but is to augment

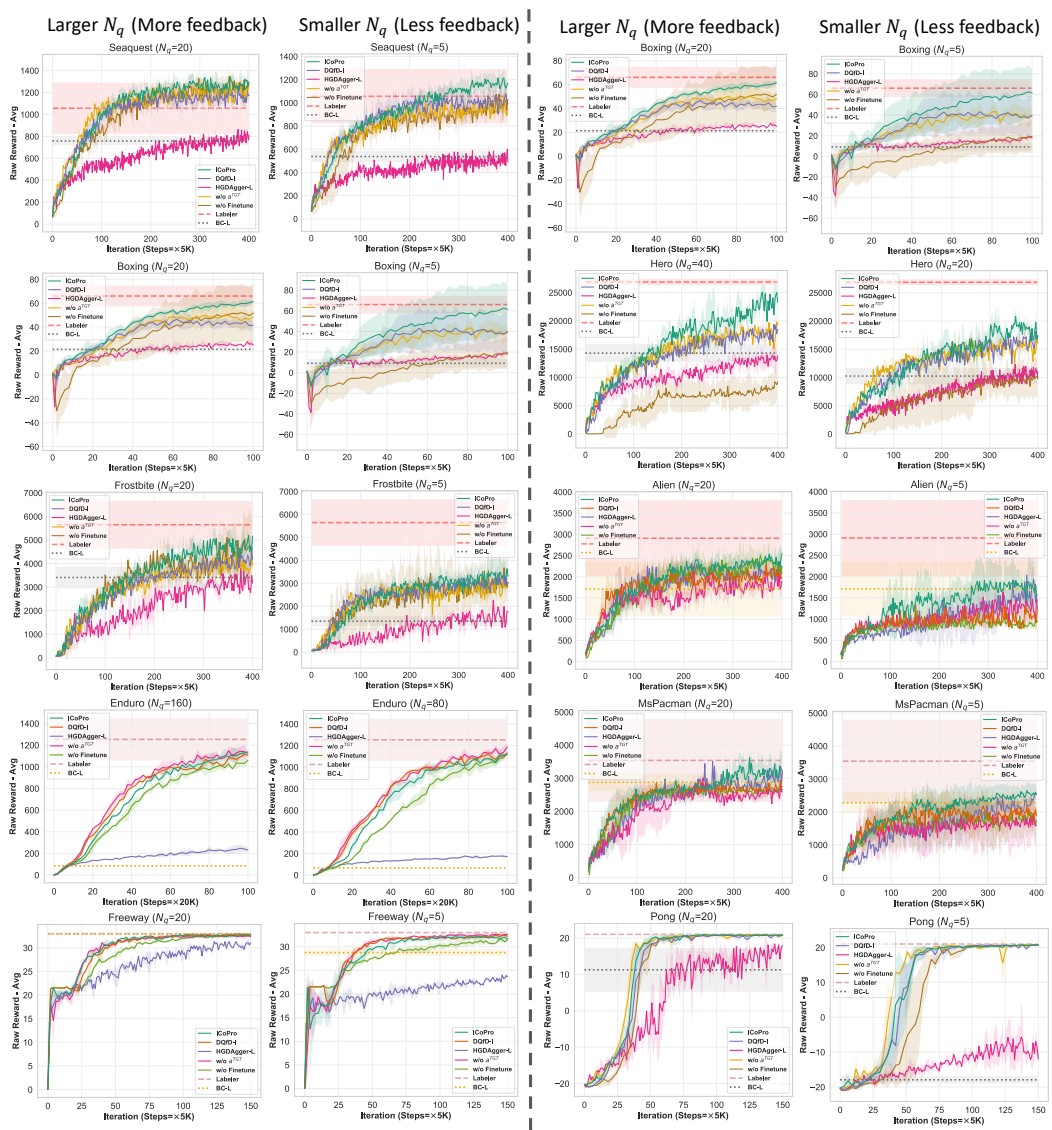

Figure 5: Ablation study on Atari to test the effect of ICoPro's two-phase scheme and target pseudo label. $N_q$ in titles refer to the number of queries per iteration, and the larger (resp. smaller) ones correspond to the large (resp. small) $\mathcal{D}^L$ in Table 2.

the replay buffer, on their experimental environments. Results for ActionImitation come from their paper. Our simulated labelers in the three games achieved the same score as theirs (full score on Pong and Freeway, and 1200 score on Enduro), so we put ICoPro's results from Figure 4 directly to compare with. Our method uses significantly less data than the ActionImitation method.

Table 3: Compare the number of feedback actions ($|N_{lab}|$) and environmental interaction timesteps ($T_{env}$) needed to reach the labelers' score in each environment.

| Method | $|N_{lab}|$ | | | $T_{env}$ | | |
|---|---|---|---|---|---|---|
| | Pong | Freeway | Enduro | Pong | Freeway | Enduro |
| ActionImitation | 10K | 10K | 10K | 3M | 3M | 3M |
| ICoPro | 375 | 2K | 8k | 375K | 500K | 2M |

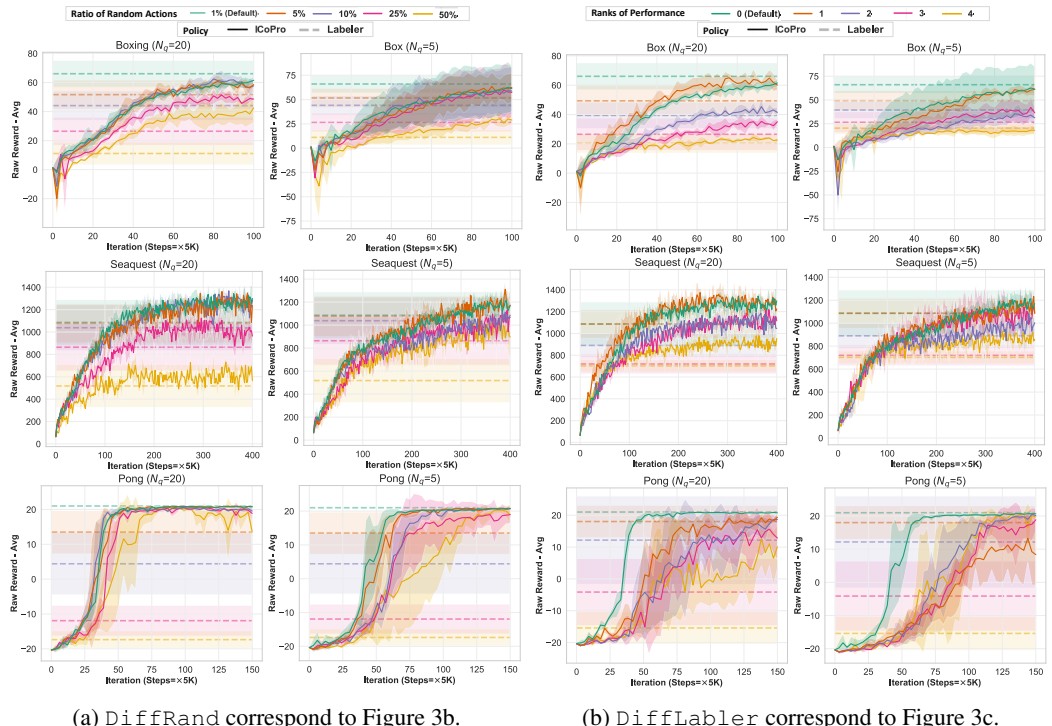

(a) `DiffRand` correspond to Figure 3b.

(b) `DiffLabler` correspond to Figure 3c.

Figure 6: Detailed plots with large/small budget size for the averaged plots in Figure 3b , although the plots for Pong are not shown in the main text due to the space limit. Larger (resp. smaller) $N_q$ indicates the large (resp. small) budget $\mathcal{D}^L$.

### A.1.3 DETAILED RESULTS FOR DIFFRAND AND DIFFLABLER WITH DIFFERENT LABEL BUDGET SIZE

Figure 6 are separated plots using different size of label busget in Figures 3b and 3c. Overall, ICoPro can overcome the two kinds of imperfections from the labeler (i.e., checkpoint) in most cases. A larger label budget tends to lead to performance with less variance in both of the two settings.

For `DiffRand`, which involves replacing part of the same labeler's labels with different ratios of random actions, a larger number of imperfect labels does not necessarily result in worse performance. This suggests ICoPro's capability to effectively utilize those good labels and overcome the effects of bad labels with proxy rewards, as illustrated in Figure 6a.

For `DiffLabler`, which employs labelers of varying performance levels, we have three main observations from the results shown in Figure 6b. (1) For ICoPro, a worse labeler does not necessarily decrease its performance. For example, in Figure 6b, the two labelers with performance ranks 0 and 1 (in green and nacarat) in Boxing-$N_q = 5$, or the two labelers with performance ranks 2 and 3 (in purple and pink) in Seaquest-$N_q = 20$, ICoPro labeled by them exhibit similar performances, despite the obvious performance gap between these two labelers. (2) Labelers with lower performance indeed tend to lead to worse performances of the trained agent from ICoPro, which is also widely noted by related works that involve human feedback (e.g., PVP (Peng et al., 2023), RLIF (Luo et al., 2024)). (3) ICoPro demonstrates a strong capability to overcome the non-optimality of the labeler in most cases: For example, in Pong, even though the worst labeler only achieves scores around -15, ICoPro trained with this labeler can still achieve scores larger than +10. But if the labeler's performance is bad, a larger number of "corrective" actions (i.e., experiments using larger $N_q$) may lead to a poorer performance of ICoPro.

## A.2 HIGHWAY

### A.2.1 EXTRA RESULTS FOR ABLATION STUDY

Figure 7 shows the ablation study on highway. For such an environment with a relatively small number of action dimensions, ICoPro can outperform HGDAgger-L significantly but only outperform other ablations slightly. Similar performance can also be observed in Atari games with small action dimension like Pong and Freeway in Figure 5.

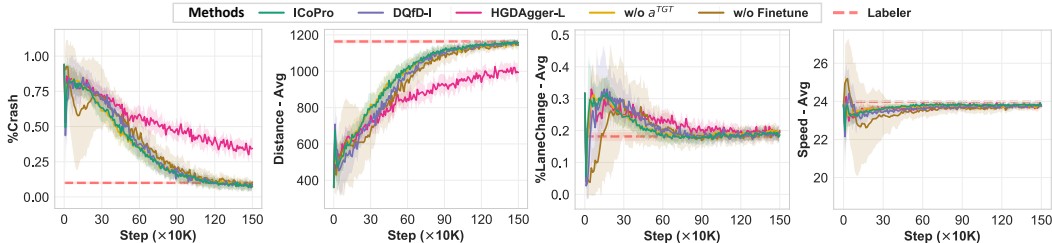

Figure 7: Extra ablation results on highway with the labeler used in Figure 2. Performances are averaged over the set of proxy rewards mentioned in Table 1.

### A.2.2 EXTRA EVALUATION WITH ANOTHER SCRIPTED LABELER (DIFFLABLER)

In Table 4, we compare the two simulated labelers used in our experiments. The first labeler (Labeler-CL), which is the labeler illustrated in our main text, prefers driving faster and changing to different lanes more often than the second one, but with a larger crash rate as well. The second one (Labeler-RL) drives in a safer way at a lower speed and prefers to drive in lanes on the right. Then, in Table 5, we list the 2 sets of proxy rewards for the two labelers.

Figure 8 shows experimental results with Labeler-RL, which confirm our analysis in Section 5.2 again that ICoPro can align to the desired performance better than baselines.

Table 4: Performance metric for the 2 simulated labelers in highway.

| Labeler | %Crash | Step-Avg | Step-Min | Distance-Avg | Distance-Min | Speed-Avg | LanePosition-Avg | %LaneChange-Avg |
|---|---|---|---|---|---|---|---|---|
| Labeler-CL | 0.10 | 48.51±0.39 | 20.80±9.78 | 1163.689±12.85 | 483.09±227.80 | 23.97±0.06 | 0.57±0.05 | 0.18±0.02 |
| Labeler-RL | 0.02 | 49.29±0.74 | 28.95±19.92 | 1099.38±19.02 | 608.93±404.97 | 22.30±0.12 | 0.91±0.02 | 0.04±0.01 |

Table 5: Reward weights for engineered proxy rewards on highway for the two labelers mentioned in Table 4.

| | Labeler-CL | | | | |
|---|---|---|---|---|---|
| | Proxy Rewards | PRExp | PR1 | PR2 | PR3 | PR4 |
| | Change lane action | 0.2 | 0 | 0.2 | 0 | 0 |
| | Normalized lane index | 0 | 0 | 0 | 0 | 0 |
| Evnets | High speed | 1.5 | 2 | 0.8 | 0 | 0 |
| | Low speed | -0.5 | -1 | 0 | 0 | 0 |
| | Crash | -1.7 | -1 | -1 | -1 | -1 |
| | Normalize | [-1,1] | [-1,1] | [-1,1] | [-1,1] | None |
| | Labeler-RL | | | | |
| | Proxy Rewards | PRExp | PR1 | PR2 | PR3 | PR4 |
| | Change lane action | 0.2 | 0 | 0 | 0 | 0 |
| | Normalized lane index | 0.5 | 0.5 | 0.2 | 0 | 0 |
| Evnets | High speed | 1.7 | 1.5 | 0.8 | 0 | 0 |
| | Low speed | -0.5 | -0.5 | 0 | 0 | 0 |
| | Crash | -1.9 | -1.5 | -1 | -1 | -1 |
| | Normalize | [-1,1] | [-1,1] | [-1,1] | [-1,1] | None |

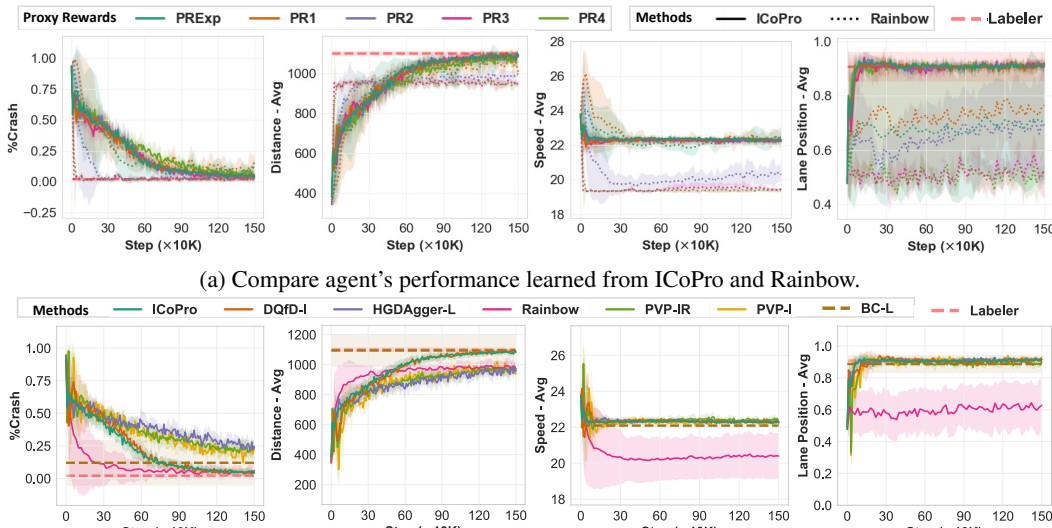

(a) Compare agent's performance learned from ICoPro and Rainbow.

(b) Compare ICoProwith other baselines. The learning curves are averaged over the set of proxy rewards mentioned in Table 5.

Figure 8: Experiments on highway with Labeler-RL using the set of proxy rewards mentioned in Table 5. Each plot compares the performance with respect to one representative performance metric. Compared with Figure 2, the different performance metric is LanePosition-Avg in the 4th subplot. $|\mathcal{D}^L|$=1.5K at the end.

### A.2.3 EXTRA RESULTS FOR DIFFRAND

In Figure 9 we show the whole detailed plots for the averaged ones mentioned in Figure 3a. ICo-Pro overcome such non-optimality inside the non-optimal corrective actions no matter which proxy reward is applied.

## B IMPLEMENTATION DETAILS FOR ICOPRO

### B.1 PSEUDO-CODE FOR ICOPRO

Algorithm 1 show the concrete learning procedure of ICoPro.

### B.2 HYPER-PARAMETERS

The basic hyper-parameters for ICoPro in all environments are listed in Table 6.

In Table 7, we list hyper-parameters with slight differences in different environments. The reasons that we do not use a same configuration is:

- Choices of $n$ in Equation (3) are the same one as training the Rainbow labeler. We find that in Boxing and Enduro we can not obtain meaningful checkpoints with the default value 20 and therefore using smaller values.

- $E$ defaults to 2, except for the 3 hard Atari games with 18 actions. We observed that setting $E = 2$ will be hard to let ICoPro and DQfD-I reach labelers' performance, but setting $E = 1$ performs better.

- Since Enduro's episode length can reach 25K when the performance approach to our labelers'. In this case, set $H$ as the default 5K can introduce bias in our iterative feedback schedule. Therefore we set $H = 20K$ in this game.

Moreover, we compare the design elements between ICoPro and the original Rainbow's (Hessel et al., 2018) in Table 8.

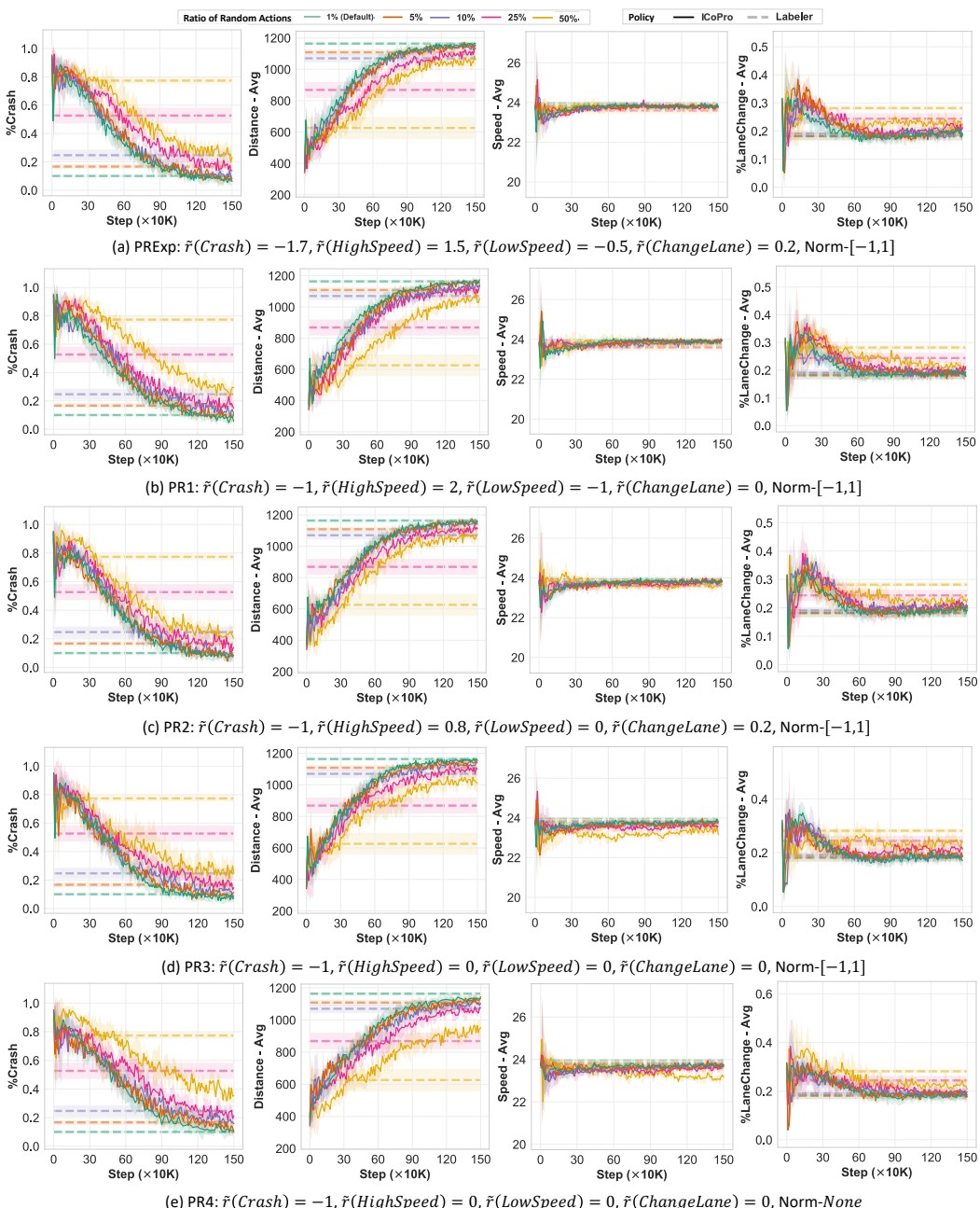

Figure 9: Detailed performance for each proxy rewards that used to show the average performance in Figure 3a.

## B.3 EXPERIMENTS COMPUTE RESOURCES

For experiments in Atari, we only need one GPU card to launch experiments, including GPUs GeForce RTX 3060 12G GPU + 48GB memory + Intel Core i7-10700F, GeForce RTX 3060 12G GPU + 64GB memory + Intel Core i7-12700, GeForce RTX 2070 SUPER + 32GB memory + Intel Core i7-9700, GeForce RTX 2060 + 64GB memory + Intel Core i7-8700. For experiments in highway, we use CPUs.

---

**Algorithm 1** ICoPro

---

**Require:** Initial policy $\pi_1^\theta$ with randomly initialized $Q_1^\theta$, oracle policy $\pi^L$ with $Q^L$
**Require:** # training iteration $N_{Itr}$, # rollout steps $H$ per iteration, # queries per iteration $N_q$
**Require:** In `finetune`-phase: Target Accuracy $\delta_{acc}$
**Require:** In `propagation`-phase: # update epochs $E$, data from the recent $I$ iterations
**Require:** $\mathcal{D}^L \leftarrow \emptyset$
  1: **for** $i \leftarrow 1$ to $N_{Itr}$ **do**
  2:    Collect a rollout $\mathcal{D}_i^{env} = \{(s_t^i, a_t^i, \widetilde{r}_t)\}_{t=1}^H$ with $a_t^i = \mathcal{G}_\epsilon(\arg\max_{a \in \mathcal{A}} Q_i^\theta(s_t^i, \cdot))$
  3:    Sample segments $\mathcal{D}_i^q = \{q_k\}_{k=1}^{N_q}$ from $\mathcal{D}_i^{env}$
  4:    **for** each $q \in \mathcal{D}_i^q$ **do**                                                        ▷ `Data Collection`-phase
  5:        Give corrective actions $\mathcal{D}_i^L = \{(s_t^i, a_{E,t}^i)_k\}_{k=1}^{N_{CF}}$
  6:        $\mathcal{D}^L \leftarrow \mathcal{D}^L \cup \mathcal{D}_i^L$
  7:    **end for**
  8:    Reset optimizer's 1st and 2nd order moment
  9:    **while** $\mathbb{E}_{s \sim \mathcal{D}^L}\left[\mathbb{I}\left[a^E = \arg\max_a Q_k^\theta(s, \cdot)\right]\right] < \delta_{acc}$ **do**                    ▷ `Finetune`-phase
 10:        Update $Q_i^\theta$ with $\mathcal{L}_L^{MG}$ (Equation (1)) on minibatch from $\mathcal{D}^L$
 11:    **end while**
 12:    Reset optimizer's 1st and 2nd order moment
 13:    **for** $j \leftarrow 1$ to $E$ **do**                                                        ▷ `Propagation`-phase
 14:        $\bar{Q} \leftarrow Q_i^\theta$
 15:        **for** minibatchs $\overline{\mathcal{D}^{env}}$ in $\mathcal{D}_{[i-I,i]}^{env}$ **do**
 16:            Calculate $\mathcal{L}_1^{RL}$, $\mathcal{L}_n^{RL}$, and $\mathcal{L}_{TGT}^{MG}$ on $\overline{\mathcal{D}^{env}}$
 17:            Calculate $\mathcal{L}_L^{MG}$ in $\overline{\mathcal{D}^L} \sim \mathcal{D}^L$, $|\overline{\mathcal{D}^L}| = |\overline{\mathcal{D}^{env}}|$
 18:            Update $Q_i^\theta$ with $\mathcal{L}^{Prop}$ (Equation (5))
 19:        **end for**
 20:    **end for**
 21:    $Q_{i+1}^\theta \leftarrow Q_i^\theta$
 22: **end for**
 23: **return** $Q_{N_{Itr}}^\theta$

---

Table 6: Default hyper-parameters.

|  | Hyper-parameter | Value |
|---|---|---|
| General | training batch size $B$ | 128 |
|  | margin $C$ | 0.05 |
|  | Architecture for Neural Network | same with DERainbow(Van Hasselt et al., 2019) |
| Optimizer | type | Adam(Diederik, 2014) |
|  | learning rate $\alpha$ | 0.0001 |
|  | eps | $0.01/B$ |
|  | betas | (0.9, 0.999) |
| `DataCollection`-phase | $N_{CF}$ | 1 |
|  | $\epsilon$ | 0.01 |
|  | $T$ | Atari: 25, highway: 10 |
|  | query sampling | uniformly sample segments without overlap |
| `Finetune`-phase | Accuracy target $\delta_{acc}$ | 0.98 |
| `Propagation`-phase | discount factor $\gamma$ | 0.99 |
|  | weight for $\mathcal{L}_{TGT}^{MG}$: $\bar{w}$ | 0.5 |
|  | weight for $\mathcal{L}_L^{MG}$: $1 - \bar{w}$ | 0.5 |

Table 7: Specific configurations and hyper-parameters for each environment. Checkpoint-step is the timestep to obtain our simulated labelers (i.e., policy checkpoints). Meanings for other configurations are consistent with Algorithm 1. The total environmental steps equals to $N_{Itr} \times H$, $|\mathcal{D}^L|$ equals to $N_{Itr} \times N_q \times N_{CF}$, $|\mathcal{D}^{env}_{[i-I,i]}|$ equals to $I \times H$.

| Configuration | Seaquest | Boxing | Battlezone | Frostbite | Atari-Games Alien | Atari-Games Hero | MsPacman | Freeway | Pong | Enduro | Highway-Labelers Labeler-CL | Highway-Labelers Labeler-RL |
|---|---|---|---|---|---|---|---|---|---|---|---|---|
| Checkpoint-step | 4.6M | 10M | 4.6M | 4.6M | 5M | 5M | 4M | 2M | 1.8M | 4.8M | 330K | 990K |
| Step $n$ | 20 | 10 | 20 | 20 | 20 | 20 | 20 | 20 | 20 | 3 | 20 | |
| $E$ | 1 | 2 | 1 | 1 | 2 | 2 | 2 | 2 | 2 | 2 | 2 | |
| $I$ | 64 | 64 | 64 | 64 | 64 | 64 | 64 | 64 | 64 | 16 | 100 | |
| $H$ | 5K | 5K | 5K | 5K | 5K | 5K | 5K | 5K | 5K | 20K | 1K | |
| $\lvert\mathcal{D}^{env}_{[i-I,i]}\rvert$ | 32K | 32K | 32K | 32K | 32K | 32K | 32K | 32K | 32K | 32K | 100K | |
| $N_{CF}$ | 1 | | | | | | | | | | 1 | |
| Small $N_q$ ($\lvert\mathcal{D}^L\rvert$) | 5 (2K) | 5 (500) | 5 (2K) | 5 (2K) | 5 (2K) | 20 (8K) | 5 (2K) | 5 (750) | 5 (750) | 80 (8k) | 10 | |
| Large $N_q$ ($\lvert\mathcal{D}^L\rvert$) | 20 (8K) | 20 (2K) | 20 (8K) | 20 (8K) | 20 (8K) | 40 (16K) | 20 (8K) | 20 (8K) | 20 (3K) | 160 (16K) | 10 | |
| $N_{Itr}$ | 400 | 100 | 400 | 400 | 400 | 400 | 400 | 150 | 150 | 100 | 150 | |
| Total Env Steps | 2M | 500k | 2M | 2M | 2M | 2M | 2M | 750K | 750K | 2M | 150K | |

## C    More Discussions About Related Works and Baselines

### C.1    Main baselines and ablations

**BC-L.**    For BC, we ran 1 seed for the replay buffer obtained at the end of training of ICoPro, therefore the performance is averaged over 5 different label buffers. In Atari, we select the best result during training with respect to during training the accuracy to reach $\delta_{acc} = 0.999$. In highway, since we have multiple performance metrics and it's not easy to measure which one is the best, we present the performance when it reaches our default $\delta_{acc} = 0.98$.

**Rainbow.**    We compare ICoPro with Rainbow in Table 8. See Table 12 for hyper-parameters of Rainbow. We use the same setting with a data-efficient version of Rainbow, except for the step $n$ and the training timesteps (see Table 7).

Table 8: Compare the different component between ICoPro and Rainbow (Hessel et al., 2018).

| Method | Components 1-step | N-step | Duel | Double Q | Replay | Distrib | Exploration |
|---|---|---|---|---|---|---|---|
| ICoPro | Yes | Yes | Yes | No | Uniform | Yes | Feedback guided |
| Rainbow | Yes | Yes | Yes | Yes | Prioritized | Yes | Noisy-net |

**Compare design choices with baselines and ablations.**    DQfD (Hester et al., 2018) is a method that learns from human demonstrations, and PVP (Peng et al., 2023) learns from human interventional control. Both of the two methods are designed by augmenting the standard RL loss with their extra IL loss into normal RL methods like DQN (Mnih et al., 2015) and TD3 (Fujimoto et al., 2018). We compare the learning scheme and corresponding losses for each baselines and ablation settings in Table 9, other hyper-parameters keep the same with Table 6 to have a fair comparison.

### C.2    Other potential baselines

#### C.2.1    RLIF

Besides baselines mentioned in the main text, RLIF (Luo et al., 2024) is another potential baseline for comparison. However, adapting RLIF's design to learn from offline corrective actions instead of the original intervention feedback leads to unsatisfactory performance. Therefore, it is not considered a primary baseline for comparison, and the results are only included in this appendix.

**Recall RLIF's method design.**    RLIF is a framework that lets humans oversee the training trajectories in real time, intervene in unsatisfactory states and take over control from unsatisfactory states to satisfactory ones. RLIF learns from these interventions by labeling the replay buffer with rewards of -1 for intervened states and 0 otherwise.

Table 9: Compare the design choices between ICoPro, different baselines, and extra ablation groups. Backslashes mean there is no such phase.

| | Methods | ICoPro | DQfD-I |
|---|---|---|---|
| Phases | Finetune | $\mathcal{L}_L^{MG}$ | \ |
| | Propagation | $0.5 \cdot \mathcal{L}_L^{MG} + 0.5 \cdot \mathcal{L}_{TGT}^{MG} + \mathcal{L}_1^{RL} + \mathcal{L}_n^{RL}$ | $\mathcal{L}_L^{MG} + \mathcal{L}_1^{RL} + \mathcal{L}_n^{RL}$ |
| | Methods | PVP-IR | PVP-I |
| Phases | Finetune | \ | \ |
| | Propagation | $\mathcal{L}_L^{PVP} + \mathcal{L}_1^{RL} + \mathcal{L}_n^{RL}$ | $\mathcal{L}_L^{PVP} + \mathcal{L}_{1(\widetilde{r}=0)}^{RL} + \mathcal{L}_{n(r=0)}^{RL}$ |
| | Methods | DAgger | w/o Finetune |
| Phases | Finetune | $\mathcal{L}_L^{MG}$ | \ |
| | Propagation | \ | $0.5 \cdot \mathcal{L}_L^{MG} + 0.5 \cdot \mathcal{L}_{TGT}^{MG} + \mathcal{L}_1^{RL} + \mathcal{L}_n^{RL}$ |
| | Methods | w/o $a^{TGT}$ | RLIF-I |
| Phases | Finetune | $\mathcal{L}_L^{MG}$ | \ |
| | Propagation | $\mathcal{L}_L^{MG} + \mathcal{L}_1^{RL} + \mathcal{L}_n^{RL}$ | $\mathcal{L}_{1(\widetilde{r}=-\mathbb{I}_c\{(s,a_\theta)\})}^{RL} + \mathcal{L}_{1(\widetilde{r}=-\mathbb{I}_c\{(s,a_\theta)\})}^{RL}$ |
| | Methods | w/o Prop's $a^L$ | w/o Prop's MG |
| Phases | Finetune | $\mathcal{L}_L^{MG}$ | $\mathcal{L}_L^{MG}$ |
| | Propagation | $\mathcal{L}_{TGT}^{MG} + \mathcal{L}_1^{RL} + \mathcal{L}_n^{RL}$ | $\mathcal{L}_1^{RL} + \mathcal{L}_n^{RL}$ |

**Experimental setups to compare with RLIF.** Although the feedback type used in RLIF is not the same as our corrective actions, we adapt their core idea (i.e., rewarding -1 for intervened states and 0 otherwise) into our framework, named RLIF-I, to have a fair comparison with ICoPro and other baselines. RLIF-I treats offline corrected states as RLIF's online intervened states. Concretely, RLIF-I is implemented by modifying ICoPro in three ways: (1) Disable the `Finetune`-phase in ICoPro, (2) Remove the margin loss used in ICoPro's `Propagation`-phase, and (3) Replacing the proxy rewards used in ICoPro's RL losses with $\widetilde{r}_{RLIF}(s,a) = -\mathbb{I}_c\{(s,a)\}$, where $a$ is the action taken by the training agent, $\mathbb{I}_c\{(s,a)\} = 1$ if $(s,a)$ has been corrected (i.e., intervened) by the labeler and 0 otherwise. We showcase this setting into Table 9 as well to provide a more clear comparison with others.

**Experimental results and analysis.** As shown in Figure 10, under the same setting as ICoPro, RLIF's learning strategy can not learn anything. This result is reasonable since the offline corrective actions setting in ICoPro is quite sparse compared with RLIF's: note that there are up to 0.8% transitions being labeled under ICoPro's labeling schedule, which can be calculated with $|\mathcal{D}^L|/TotalEnvSteps$ in Table 7, while RLIF requires to intervene all unsatisfied states. Such results indicate that making RLIF work requires intervening in many more states than ICoPro, which is too label-intensive to be scaled for more complex and practical tasks.

## C.3 OTHER POTENTIAL ABLATIONS

### C.3.1 EXTRA ABLATIONS ON MARGIN LOSSES IN THE `Propagation`-PHASE

For the margin losses with $a^L$ (Equation (1)) and $a^{TGT}$ (Equation (4)) used in our `propagation`-phase, we mainly focus on the ablation that removing Equation (4) with $a^{TGT}$ but keeping Equation (1) with $a^L$ in our main text. As discussed in Section 4, removing Equation (1) from the `propagation`-phase is problematic since the corrective actions in $\mathcal{D}^L$ are not executed in $\mathcal{D}^{env}$; that is, RL losses (Equations (2) and (3)) are optimized with off-policy trajectories and the newly provided corrective actions $a^L$ from the labeler are never taken in those trajectories.

To make the ablation study more thorough, we show two extra ablations related to the margin loss with $a^L$ used in the `propagation`-phase: (1) removing $a^L$ but keeping $a^{TGT}$ (denoted as w/o Prop's $a^L$), or (2) removing both $a^L$ and $a^{TGT}$ (denoted as w/o Prop's MG). Table 9 compares the two ablation settings with other experimental ablations or baselines. As shown in Figure 11, both the three ablations perform worse than ICoPro, indicating the necessity of keeping the margin loss with both $a^L$ and $a^{TGT}$ in the `propagation`-phase.

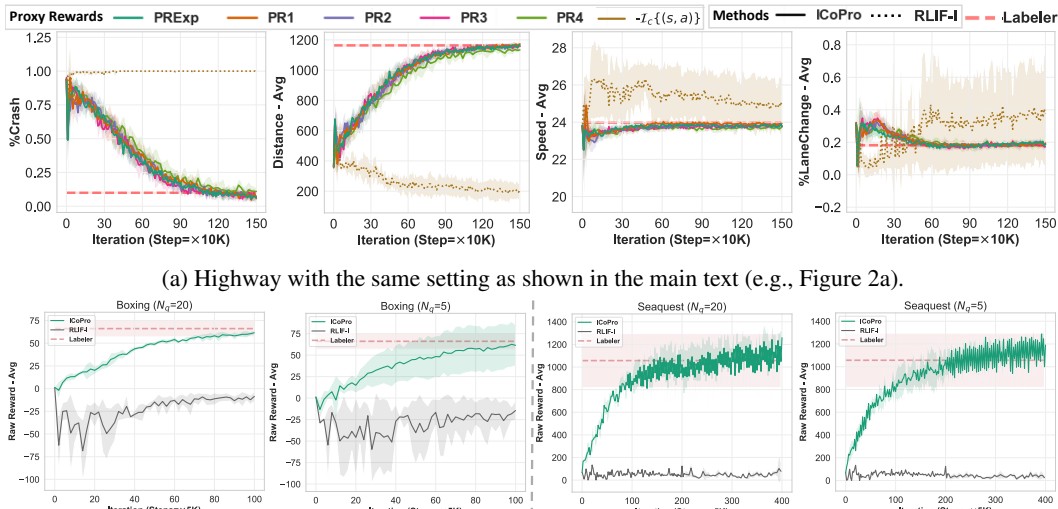

(a) Highway with the same setting as shown in the main text (e.g., Figure 2a).

(b) Two Atari games (Boxing and Seaquest) with the same setting as shown in the main text (e.g., Table 2).

Figure 10: Compare RLIF-I with ICoPro under the same setting (e.g., simulated labeler, feedback schedule and budget).

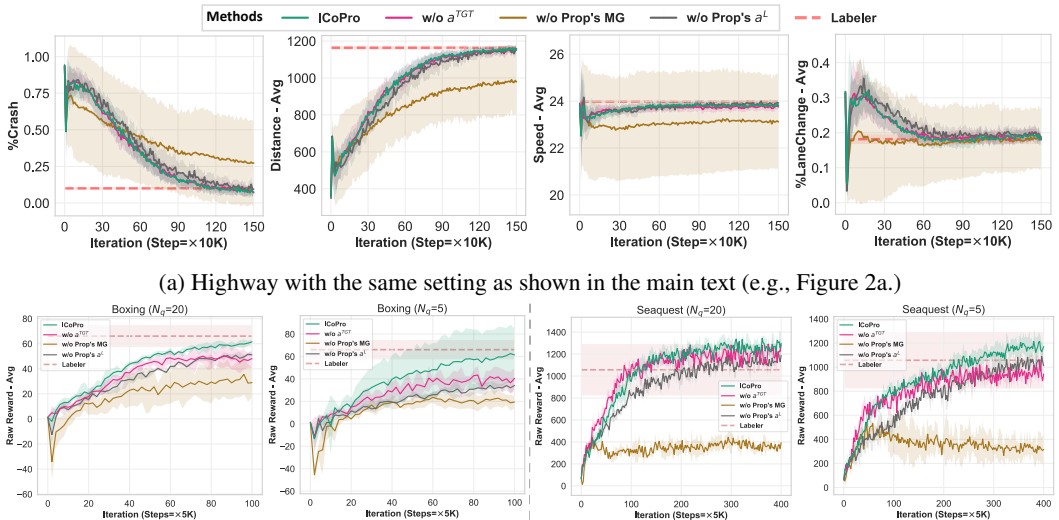

(a) Highway with the same setting as shown in the main text (e.g., Figure 2a.)

(b) Two Atari games (Boxing and Seaquest) with the same setting as shown in the main text (e.g., Table 2).

Figure 11: Ablations with respect to the margin losses with $a^L$ and $a^{TGT}$ in the `propagation`-phase. Compared with the default setting of ICoPro's `propagation`-phase, w/o $a^{TGT}$ refers to removing $a^{TGT}$ but keeping $a^L$, w/o Prop's $a^L$ refers to removing $a^L$ but keeping $a^{TGT}$, w/o Prop's MG refers to removing both $a^L$ and $a^{TGT}$. Note that as one of our main ablation settings, w/o $a^{TGT}$ has been tested besides

# D    MORE DETAILS ABOUT EXPERIMENTAL SETTINGS

## D.1    CONFIGURATIONS FOR EVALUATION

We use 5 seeds for ICoPro and baselines for evaluation, and 3 seeds for ablations. For all experiments we perform, we use 50 episodes in each evaluation. When we ran experiments for ICoPro and baselines, we used 5 seeds for each setting. For ablation studies, we use 3 seeds. Although ICoPro

and its' ablation w/o `Finetune` involve 2 separate phase in each iteration, their performance are still evaluated at the end of each iteration, which is the `propagation`-phase.

## D.2 HIGHWAY

Highway (Leurent, 2018) is an environment with state-based inputs in 35 dimensions and 5 available action choices to control the vehicle's speed and direction. The input states contain information about both the controlled vehicles and the top 5 nearest vehicles' positions, speeds, and directions. The basic goal in this environment is to drive a car on a straight road with multiple lanes within a limited time. One episode is terminated when the time is used up, or a crash happens.

We use the configuration listed in Table 10 in highway.

Table 10: Hyper-parameters for environment in highway.

| | Hyper-parameter | Value |
|---|---|---|
| Env | Available speed range | [19,30] |
| | High speed range | $\geq 21$ |
| | Low speed range | $< 21$ |
| | Lanes count | 5 |
| | Vehicles count | 40 |
| | Time limit | 50 |
| | Policy frequency | 1 |
| Observation | Type | Kinematics matrix |
| | Number of observed vehicles | 5 |
| | Features for each vehicle | [presence, x, y, $v_x$, $v_y$, $\cos_{heading}$, $\sin_{heading}$] |
| Action | Type | [Right, Left, Faster, Slower, IDLE] |

## D.3 ATARI

In Atari, we use the signed raw reward from the environment as the proxy reward in our setting: $\widetilde{r}_A = sign(r_A)$, where $r_A$ is the raw reward in Atari and $\widetilde{r}_A$ is the proxy reward. The average cumulative $r_A$ of episodes serves as a performance metric.

Both $r_A$ and $\widetilde{r}_A$ have different levels of imperfection. The imperfection of $\widetilde{r}_A$ mainly comes from two aspects: (1) Missing rewards, e.g. in Seaquest, $r_A(LooseLives) = 0$, but give a negative reward is better; (2) Numerical mismatch between $\widetilde{r}_A$ and $r_A$, e.g., in Hero, $r_A(RescueMiners) = 1K$, $r_A(ShootCritters) = 50$ but $\widetilde{r}_A = 1$ for both the two cases. More information about the ground-truth $r_A$ can be checked in the official website[6].

Our environment configurations follow standard requirements Machado et al. (2018). Table 11 lists the related details.

Table 11: Hyper-parameters for the Atari environment wrapper.

| Hyper-parameter | Value |
|---|---|
| Grey-scaling | True |
| Observation down-sampling | (84, 84) |
| Frame stacked | 4 |
| Frame skipped | 4 |
| Action repetitions | 4 |
| Max start no ops | 30 |
| Reward clipping | [-1, 1] |
| Terminal on loss of life | True |
| Max frames per episode | 108K |

---

[6]https://gymnasium.farama.org/environments/atari/

### D.4 SIMULATED LABELERS

We use Rainbow to train labelers for all environments, with the general hyper-parameters shown in Table 12 and concrete training timesteps in Table 7. For highway, script labelers are trained with PRExp mentioned in Tables 1 and 5. For Atari, script labelers are trained with the signed raw rewards. Note that these script labelers are not necessarily perfect ones since their performance could be improved further with longer training.

Table 12: Hyper-parameters for Rainbow.

| Hyper-parameter | Value |
| --- | --- |
| Update period for target network | per 2000 updates |
| Atoms of distribution | 51 |
| $\gamma$ | 0.99 |
| Batch size | 32 |
| Optimizer | Same with Table 6 |
| Max gradient norm | 10 |
| Prioritized replay | exponent: 0.5, correction: 0.4→1 |
| Noisy nets parameter | 0.1 |
| Warm-up steps | 1.6K |
| Replay buffer size | 1M |
| step $n$ | Same with Table 7 |
| Sample steps to update | 1 |
| Netowrk-Atari | CNN encoder: channels [32, 64], kernel size & strides [5, 5], paddings [3, 1] MLP policy: hidden size [128] |
| Netowrk-Highway | MLP encoder: hidden size [128, 128] MLP policy: hidden size [128] |

## E MORE DETAILS ABOUT USER STUDY

Appendix E.1 and Appendix E.2 demonstrate the detailed evaluation results of ICoPro-Human. Then Appendix E.3 shows some examples of the corrective actions provided by humans. Finally, Appendix E.4 gives an introduction of tasks and corresponding user interface.

### E.1 VISUALIZATION FOR AGENTS TRAINED WITH ICOPRO-HUMAN

In Figure 12 we visualize an action segment from Pong to validate the human-like performance trained by ICoPro-Human comparing with the *scripted labeler*.

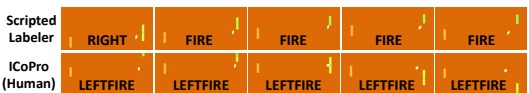

Figure 12: Performances of the *scripted labeler* and *ICoPro-Human* in Pong. Each row shows a sequence of state-action pairs.

Other evaluation videos for ICoPro-Human are sorted in `SupplementaryMaterials/Videos [Env]UserStudy`, which learns to perform more like human players compared with evaluation videos shown in `SupplementaryMaterials/Videos[Env]Oracles`.

### E.2 TRAINING PLOTS

Figure 13 shows some statistics of ICoPro-Human during training.

- For highway, we show the two related performance metrics (i.e., $\%Crash$ and $Speed - Avg$) following the instruction described in Table 15.

- For the two Atari games, we can only show their game scores. However, the intentions described in Table 13 are not entirely consistent with the game scores, as the human labeler aims to teach additional performance aspects that are not captured by the proxy rewards.

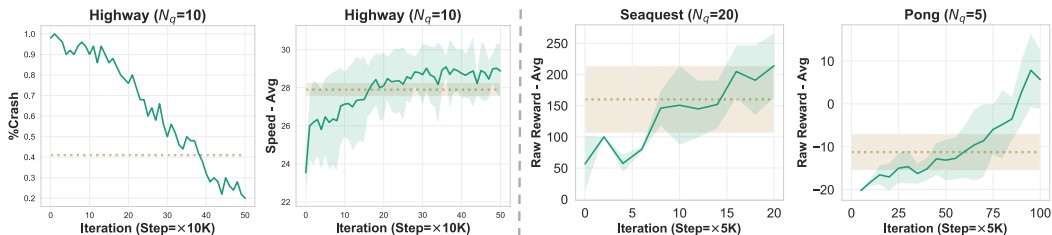

Figure 13: Performance of ICoPro-Human during training. Solid lines (resp. dashed lines) are the performance of ICoPro-Human (resp. BC-L), with shadows indicating the standard deviation measured over 50 episodes.

### E.3 EXAMPLES OF HUMAN CORRECTIVE ACTIONS

In Figure 14, we showcase some visualizations of the states that human labelers selected, the original agent actions, and the provided corrective actions. Contextual information (i.e., what happens before and after a state) is available to the human labelers (e.g., with our simple interface described in Figure 15) to help them select state(s) and provide corrective action(s). For example:

- In Pong (Figure 14a): human labelers can judge if an agent action is reasonable or not by knowing the movement of the ball.

- In Seaquest: the left-most example of Figure 14b shows that human labelers can determine which state to correct by assessing whether the agent's actions will result in a fire hitting a fish. The other three examples of Figure 14b demonstrate that human labelers can effectively guide the agent by knowing the movement of fish and divers before a state.

- In Highway: the left two examples of Figure 14c show that human labellers can provide corrective actions by knowing the movement of cars, while the right two examples show that human labelers can provide corrective actions by knowing the consequences (e.g., car crash in the two examples) of agent actions on those states.

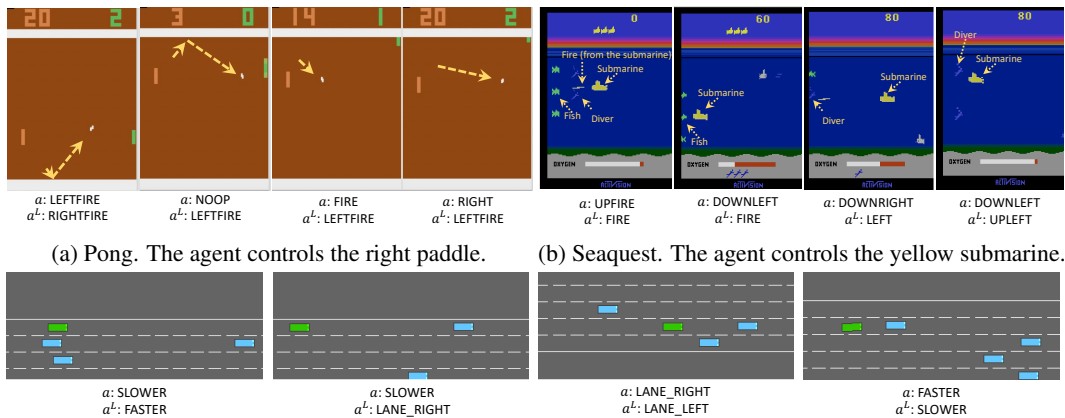

(a) Pong. The agent controls the right paddle.    (b) Seaquest. The agent controls the yellow submarine.

(c) Highway. Cars are driving from left to right. The agent controls the green car.

Figure 14: Examples of human actions provided for ICoPro-Human. Each image indicates a state, while $a$ and $a^L$ refer to the agent action and human action in that state, respectively. In Figure 14a, we add yellow dashed arrows to denote the movement direction of the ball to avoid confusion. In Figure 14b, we add yellow text and dashed arrows to denote the role of objects.

### E.4 PROCEDURE OF USER STUDY

No need to let humans play the game by themselves before providing feed-back. Before letting the human labelers give feedback, we show them an instruction about the task they will participate (see Tables 13 and 15), as well as an example video to let them become familiar with the environment dynamics (see `PATH/TO/SupplementaryMaterial]/Video[EnvName]Oracles-Examples`).

We show our simple interface used in our user study in Figure 15.

Table 13: User instruction for Pong.

---

**User Instructions - Pong**

**Description**:
- You control the right paddle, you compete against the left paddle.
- You each try to keep deflecting the ball away from your goal and into your opponent's goal.

- - - - - - - - - - - - - - - - - - - - - - - - - - - - - - - - - - - - - - - - - -

**Reward**:
- A player scores +1 when the opponent hits the ball out of bounds or misses a hit, or -1 when you hits the ball out of bounds or misses a hit.
- The first player or team to score 21 points wins the game.

- - - - - - - - - - - - - - - - - - - - - - - - - - - - - - - - - - - - - - - - - -

**Actions**:

| Value | Meaning | Value | Meaning | Value | Meaning |
|-------|---------|-------|-----------|-------|----------|
| 0 | NOOP | 1 | FIRE | 2 | RIGHT |
| 3 | LEFT | 4 | RIGHTFIRE | 5 | LEFTFIRE |

- `NOOP`: random action
- `LEFT`: move your paddle down
- `RIGHT`: move your paddle up
- `FIRE`: add some speed to the return ball or put sharper angles on your return hits when the ball contacts with your paddle
- Other actions are combined effects as described above.

---

Table 14: User instruction for Seaquest.

---

**User Instructions - Seaquest**

**Description**:

- You control a submarine to (1) save divers, and (2) shoot fish.
- Pay attention to the oxygen bar: you must float to the sea surface before using up your oxygen, otherwise, you will lose a life.
    - Each time you surface, you must bring at least one diver with you, or you will lose a life.
    - Get in touch with the divers then they will enter your submarine.
- Pay attention to the fish: you can shoot them, but if you touch them or are hit by their fire, you will lose a life.
- You have three lives in total.

- - - - - - - - - - - - - - - - - - - - - - - - - - - - - - - - - - - - - - - - - - - - -

**Reward**:

- +1 for shooting a fish and 0 otherwise

- - - - - - - - - - - - - - - - - - - - - - - - - - - - - - - - - - - - - - - - - - - - -

**Actions**:

- `NOOP`: random action
- `LEFT` / `RIGHT` / `UP` / `DOWN`: control the frontal orientation of the submarine
- `FIRE`: shoot a fire from the front of the submarine
- Other actions (`UPRIGHT` / `UPLEFT` / `DOWNRIGHT` / `DOWNLEFT` / `DOWNRIGHT` / `UPFIRE` / `DOWNFIRE` / `LEFTFIRE` / `RIGHTFIRE` / `UPRIGHTFIRE` / `UPLEFTFIRE` / `DOWNRIGHTFIRE` / `DOWNLEFTFIRE`) are combined effects as described above.

---

Table 15: User instruction for highway.

---

**User Instructions - highway**

**Description**:

- You control the green vehicle. There are other blue vehicles around.
- You need to drive at a speed as large as you can while avoiding a crash.
- You need to take over other vehicles if possible.

- - - - - - - - - - - - - - - - - - - - - - - - - - - - - - - - - - - - - - - - - - - - -

**Reward**:

- -1 for collision and 0 otherwise

- - - - - - - - - - - - - - - - - - - - - - - - - - - - - - - - - - - - - - - - - - - - -

**Actions**:

- `LANE_LEFT`: Change lane to the left, no effect if already in the leftmost lane.
- `LANE_RIGHT`: Change lane to the right, no effect if already in the rightmost lane.
- `IDLE`: No action, keep the current speed and heading direction.
- `FASTER`: Faster.
- `SLOWER`: Slower.

---

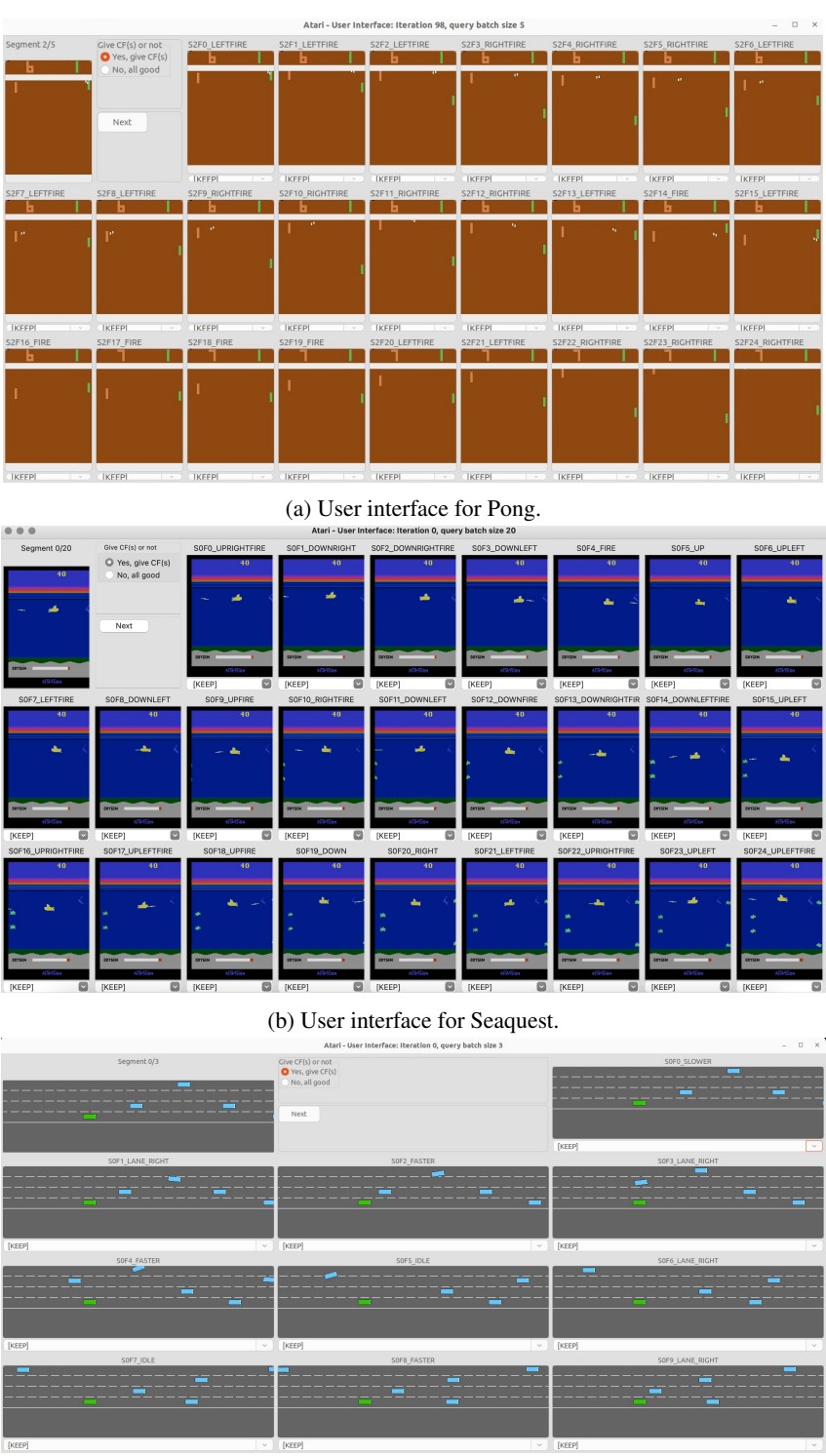

(a) User interface for Pong.

(b) User interface for Seaquest.

(c) User interface for highway.

Figure 15: User Interface. We put the video of that segment in the top left corner. If the labeler is satisfied with the whole segment's state-action pairs, they can choose to pass this segment with the radio box at the top near the video window. Otherwise the labeler will give corrective actions on other windows.

