# OpenReview forum: "Reinforcement Learning from Imperfect Corrective Actions and Proxy Rewards"
_ICLR.cc/2025/Conference — ICLR 2025 Poster_

### Official Review · Reviewer_kqSo · 2024-10-28

**Soundness:** 3
**Presentation:** 3
**Contribution:** 3
**Rating:** 8
**Confidence:** 5

**Summary:**

This paper introduces a novel method that integrates additional (imperfect) corrective actions into the reinforcement learning (RL) process to address limitations in the imperfect proxy reward function.

**Strengths:**

The paper is well-organized and easy to follow, and the experimental section provides sufficient evidence to support the authors' claim that their approach improves the learning of an optimal policy.

**Weaknesses:**

1. The reviewer is curious about the performance information available to the labeler that allows them to decide whether an action is good or bad, considering that the queries are sampled from a dataset rather than occurring in real-time. In the experiments, a trained Q-network is used as the labeler, which simplifies the collection of corrective actions. Although there is a user study involving human labelers in two environments (Highway and Pong), this does not fully address my concern.

2. The term "sample-efficient" is mentioned frequently throughout the paper, yet no clear metrics are provided to quantify this efficiency. It would be helpful to include some metrics to validate this claim.

3. Some relevant literature is missing from the discussion. For instance, ranking-based reinforcement learning [1]. The authors could also refer [2], which might offer more effective corrective actions than relying solely on the target Q-network, especially during the early stages when the model is undertrained.

There are a few minor issues with the writing. For example:

In the related work section, the discussion on the limitations of the offline setting is mentioned but not clearly explained. Providing more clarity here would be helpful. Table I includes some abbreviations, such as PR1 and PR2, which I assume stand for Proxy Reward. However, additional explanation would improve understanding for readers unfamiliar with these terms.

Reference:
[1] Brown, Daniel S., Wonjoon Goo, and Scott Niekum. "Better-than-demonstrator imitation learning via automatically-ranked demonstrations." Conference on robot learning. PMLR, 2020.
[2] Zhan, Huixin, Feng Tao, and Yongcan Cao. "Human-guided robot behavior learning: A gan-assisted preference-based reinforcement learning approach." IEEE Robotics and Automation Letters 6.2 (2021): 3545-3552.

**Questions:**

1. What interface is used for the corrective action process?
2. The entire approach is based on the Q-function, which is limited to discrete action spaces. Can this method be extended to continuous action spaces (without discretization)?

---

> ### Author Response · Authors · 2024-11-20
> **Thank you for you feedback, and about Weakness 1-3 and Question 1**
>
> Thank you for your constructive insights and meticulous feedback!
>
> **Here we first recall the review comments with quotes, then give our feedback. For the mentioned updates to pdf below, we use the blue color in the latest paper pdf to denote updated content.**
>
> We sincerely hope that our response clearly addresses the raised concerns, and we are more than willing to discuss any remaining questions.
>
> # W1 & Q1: About contextual information to let a human give corrective actions
>
> >Q1: What interface is used for the corrective action process?
>
> > W1: The reviewer is curious about the performance information available to the labeler that allows them to decide whether an action is good or bad, considering that the queries are sampled from a dataset rather than occurring in real-time. In the experiments, a trained Q-network is used as the labeler, which simplifies the collection of corrective actions. Although there is a user study involving human labelers in two environments (Highway and Pong), this does not fully address my concern.
>
> - **In Figure 11 in the original manuscript (Figure 15 in the newly updated pdf), we show our interface used for user study**. Inside the interface, we have three kinds of sub-windows:
>   - (1). One sub-window to display the video of the queried segment, allowing labelers to quickly understand the context.
>   - (2). One sub-window to let the labeler decide if s/he provides corrective action(s) for this segment,
>   - (3). And other sub-windows to show the specific frames and agent actions for concrete timesteps, along with a pull-down list in these sub-windows to enable the labeler to provide the correct action on the selected state(s).
> - Therefore, **contextual information (i.e., what happens before and after a state) is available to the human labeler to help them select state(s) and provide corrective action(s)**. We demonstrate some examples of corrective actions provided by human labelers, in our updated manuscript's Appendix E.3.
> - Moreover, **we choose another more complex Atari game Seaquest, with a larger action space ($|\mathcal{A}|$=18)  than Highway ($|\mathcal{A}|$=5) and Pong ($|\mathcal{A}|$=6), to further enrich our experiments with real humans**. If interested, the reviewer can further check the results of Seaquest in Section 5.4 and Appendix E, which verifies again ICoPro-Human's ability and efficiency in obtaining the human-expected policy.
>
> # W2: About the definition of "sample-efficient"
> > W2: The term "sample-efficient" is mentioned frequently throughout the paper, yet no clear metrics are provided to quantify this efficiency. It would be helpful to include some metrics to validate this claim.
>
> **"Sample-efficient" refers to both environmental transitions and corrective actions**, as Lines 77-78 explain.
> We apologize for causing the reviewer's confusion about this point, due to our use of "data-efficient" instead of "sample-efficient" in Lines 77-78 in our original manuscript. We have corrected this word in the updated manuscript.
>
> # W3: About missing relevant literature
> > Some relevant literature is missing from the discussion. For instance, ranking-based reinforcement learning [1]. The authors could also refer [2], which might offer more effective corrective actions than relying solely on the target Q-network, especially during the early stages when the model is undertrained.
> Reference:
> [1] Brown, Daniel S., Wonjoon Goo, and Scott Niekum. "Better-than-demonstrator imitation learning via automatically-ranked demonstrations." Conference on robot learning. PMLR, 2020.
> [2] Zhan, Huixin, Feng Tao, and Yongcan Cao. "Human-guided robot behavior learning: A gan-assisted preference-based reinforcement learning approach." IEEE Robotics and Automation Letters 6.2 (2021): 3545-3552.
>
> We sincerely thank the reviewer for sharing the two interesting papers, and thinking about potential improvements for our work.
> - In our new manuscript, **we now cite both references** in the cluster of work under the umbrella of "preferences", including pairwise comparison to ranks (Line 107). For [1], we also cite it in Line 139 when talking about inverse RL.
> - For [2], we don't think it would work efficiently in our case.
>   - In short, in their case, predicting human feedback (i.e., pairwise comparison for segments) can be modeled in two formats: training model A (i.e., a binary preference) costs less human feedback, while model B (i.e., a sequence of predicted rewards) is required for policy training. Then [2] uses A as a human assistant to provide fake feedback for training B.
>   - Although not completely impossible, currently we don't think it's natural to transfer such a design from the *pairwise segment comparison* setting to our *corrective action* setting.

---

> > ### Comment · Reviewer_kqSo · 2024-11-24
> >
> > Thank you to the authors for addressing the missing relevant literature. If Literature 2 is not relevant to the current methodology, it may be excluded.

---

> ### Author Response · Authors · 2024-11-20
> **About weakness 4 and question 2**
>
> # W4: Suggestions for writing
> > There are a few minor issues with the writing. For example:
> > 1. In the related work section, the discussion on the limitations of the offline setting is mentioned but not clearly explained. Providing more clarity here would be helpful.
>
> Thank you for your careful reading, we agree with this comment and **have updated that paragraph by explaining the limitations of the offline setting more clearly in the new manuscript**.
>
> > 2. Table I includes some abbreviations, such as PR1 and PR2, which I assume stand for Proxy Reward. However, additional explanation would improve understanding for readers unfamiliar with these terms.
>
> Yes, we use "PR" as an abbreviation for "Proxy Reward". **The set of proxy rewards is explained in Table 1**, which is a set of "representative proxy rewards in different levels of imperfection by associating events with different rewards" as introduced in Lines 344-346.
> We apologize for causing the reviewer's misunderstanding of this point, and we have improved the caption of Figure 2 by mentioning Table 1 there in our updated manuscript.
>
> # Q2: Potential extension of our method to continuous space, without discretization
> > The entire approach is based on the Q-function, which is limited to discrete action spaces. Can this method be extended to continuous action spaces (without discretization)?
>
> **Extending our method to continuous action space is possible by instantiating our method with some actor-critic architectures (e.g., TD3[5]), but other imitation learning losses should be considered instead of margin losses (eq.1 and eq.4).**
> Since the margin loss is only applicable in the discrete action setting, instead we could consider, **for instance, the following two candidates**:
> - 1. Enforce a predefined upper bound $B_{>0}$ for the Q values, and push $Q^{\theta}(s, a^L)$ to that bound, which is adapted from
>       - PVP [6] (e.g., $\mathcal{L}^B(\theta|\mathcal{D}^L)=\mathbb{E}_{(s,a^{L})\in\mathcal{D}^L}\left[|Q^{\theta}(s,a)-B|^2\right]$) ,
>       - or [7] (e.g., $\mathcal{L}^B(\theta|\mathcal{D}^L)=\mathbb{E}_{(s,a^L)\in\mathcal{D}^L}\left[max(0, Q^{\theta}(s,a)-B)\right]$).
>
>     - Use $\mathcal{L}^B+\mathcal{L}^{TD}$ to train $Q^{\theta}$, where $\mathcal{L}^{TD}$ refers to the default RL losses.
>     - No need to modify the actor's training part. The actor is trained by maximizing $Q^{\theta}$.
> - 2. Penalize $Q(s,a)$ based on a distance measure between $a$ and $a^L$ when optimizing the policy, which is adapted from methods like [8].
>   - Only use $\mathcal{L}^{TD}$ to optimize $Q^{\theta}$
>   - $\pi=\arg\max_{\pi}\mathbb{E}_{(s,a^L)\sim\mathcal{D}^L}\left[\lambda Q^{\theta}(s,\pi(s))-(\pi(s)-a^L)^2\right]$
>
> # Mentioned References
> To avoid confusion with the references listed by the reviewer, the index starts from 3.
>
> [3] Ho, Jonathan, and Stefano Ermon. "Generative adversarial imitation learning." Advances in neural information processing systems 29 (2016).
>
> [4] Reddy, Siddharth, Anca D. Dragan, and Sergey Levine. "SQIL: Imitation Learning via Reinforcement Learning with Sparse Rewards." International Conference on Learning Representations.
>
> [5] Fujimoto, Scott, Herke Hoof, and David Meger. "Addressing function approximation error in actor-critic methods." International conference on machine learning. PMLR, 2018.
>
> [6] Peng, Zhenghao Mark, et al. "Learning from active human involvement through proxy value propagation." Advances in neural information processing systems 36 (2024).
>
> [7] Spencer, Jonathan, et al. "Learning from interventions: Human-robot interaction as both explicit and implicit feedback." 16th Robotics: Science and Systems, RSS 2020. MIT Press Journals, 2020.
>
> [8] Fujimoto, Scott, and Shixiang Shane Gu. "A minimalist approach to offline reinforcement learning." Advances in neural information processing systems 34 (2021): 20132-20145.

---

> > ### Comment · Reviewer_kqSo · 2024-11-24
> >
> > Thank you for the revisions to the paper. It looks good to me, and I have adjusted my score accordingly.

---

> > > ### Author Response · Authors · 2024-11-25
> > >
> > > Thank you for raising the score. We are pleased that the quality of this work has been improved thanks to your insightful questions and suggestions!

---

### Official Review · Reviewer_QvY8 · 2024-10-29

**Soundness:** 3
**Presentation:** 3
**Contribution:** 3
**Rating:** 8
**Confidence:** 4

**Summary:**

This paper proposes to use proxy reward (the reward that is easily defined and might not accurately reflects the desired behavior) and the offline corrective action (labeller assigns correct actions to the rollouts generated by the agent, and those corrective actions will not be used to the environment as the labelling process is offline) to train policies.

The training can be split into 3 phases: data collection, finetune-phase, and propagation-phase. In the data collection phase, agent rolls out trajectories with epsilon-greedy exploration and human provide offline corrective actions. In finetune phase, marginal loss is applied to the Q values to make sure the Q values for corrective actions are high. In propagation phase, RL loss (TD loss with the proxy reward) and the marginal loss (the imitation loss that enforces the learned policy to be closed with previous agent policy) are used together.

**Strengths:**

1. Good to see the author provides source code.
2. Human-in-the-loop with offline corrective feedback is a quite interesting setting. Previous work either focus on online demonstration (HGDagger, PVP etc) or offline labeling (PbRL, TAMER), but this work focus on offline corrective action (not the reward labeling but the action).
3. Human-in-the-loop methods suffer from the cost of human subjects, as they need to be very attentive if in a online human-AI shared control setting. This paper adopts the offline setting and utilize the proxy reward, which is an interesting way to scale up the human-in-the-loop method.

**Weaknesses:**

1. Further analysis will greatly improve the technical depth of the paper. Further analysis on ablations and why the method work is needed. These include: why marginal loss in propagation phase works? How well the method work with human labeler in different optimality? The ablations study are presented in Section 5.2 and Table 2 while in Appendix A, while there is no discussion on the reasons. The author just put the results there. See more on "Questions".
2. All ablation studies are conducted in the Atari, and further discussion is needed. Why ablations fail? What insights can we get from the ablation studies?
3. As a paper studying human-in-the-loop policy learning, there is no enough weights on real-human experiments. Section 5.4 shows the experiment with real human subject. I will expecting far more information about this experiment. Why don't put the evaluation results of the trained policies in each iteration of the training and compare it against the baselines? Is there qualitative results showing the human's corrective actions and the agent's actions?
4. Follow question 3, providing corrective actions is actually very boring and time-consuming. The setting might be even more exhausting than online human supervision. I doubt whether the method is applicable at all in real-human setting.

**Questions:**

1. Why the marginal loss in propagation phase works? IIUC the "w/o aTGT" ablation is the experiment ablating the marginal loss in propagation correct? It seems that adding that term can help in terms of the final results. But more analysis is needed. For example, will adding the term improve the learning efficiency and reduce human interventions? In Sec 5.2 the ablation studies only show the performance in tasks but the other metrics such as "human efficiency" is not compared.
2. How to ensure the label quality? In more complex task like driving, human labeler is hard to understand the outcome of their "corrective actions". For example to rescue a car from crashing left car, the human labeler might steer the car all to the right hand side. And this action though seems to be correct, but it's impossible to be applied in real world (you can't rotate the steering wheel immediately from left to right), and it will also cause unknown outcome such as car rollover. There is no guarantee on the quality of the data especially in the offline setting. In motivation the authors suggest this paper can work with suboptimal human labeler but **no experiment shows the method still work on suboptimal experts**.
3. I will expect more discussion on human cost. What designs in the method can improve human efficiency (less corrective actions)? Line 259-261 says "In addition, training with pseudo-labels can
reduce the cost of collecting human labels by leveraging the large number of unlabeled states in
Denv" No experiment supports this claim and I will consider this to be a overclaim. Is there any evidence showing that the agent converge to same performance with less human (here the simulated expert) involvement?

---

> ### Author Response · Authors · 2024-11-18
> **Thank you for your feedback**
>
> Thank you for your constructive insights and meticulous feedback! We are deeply grateful for the reviewer's recognition that our framework has interesting potential effectiveness compared with existing methods.
>
> **Here we first recall the review comments with quotes, then give our feedback. For the mentioned updates to pdf below, we use the blue color in the latest paper pdf to denote updated content.**
>
> We sincerely hope that our response addresses the concerns raised, and we are more than willing to discuss them if there are any remaining questions.

---

> ### Author Response · Authors · 2024-11-18
> **About W1-2 & Q1-3  (Part1)**
>
> # W1 & W2 & Q1 & Q2 & Q3: About margin loss in the propagation phase, ablation studies, human cost, and experiments with non-optimal labelers
> **Considering their internal dependency, we merge and classify W1-2 and Q1-3 into four points. Then for each point, we use quote format to cite related comments from the reviewer**.
> ## 1. About margin loss in the propagation phase
> > [1.1]. Why does marginal loss in the propagation phase work?
>
> > [1.2]. (About "w/o a^{TGT}") It seems that adding that term can help in terms of the final results. But more analysis is needed. For example, will adding the term improve the learning efficiency and reduce human interventions?
>
> > [1.3]. Line 264-266 says "In addition, training with pseudo-labels can reduce the cost of collecting human labels by leveraging the large number of unlabeled states in Denv" No experiment supports this claim and I will consider this to be a overclaim.
>
> Since the margin loss in the propagation phase of ICoPro involves both $a^{L}$ and $a^{TGT}$ (eq.1 and eq.4), we give feedback on the above three questions together.
>
> - **Methodological analysis**
>
>   Here we analyse the effect of $a^L$ and $a^{TGT}$ one by one:
>   - (1) Necessity of the margin loss with $a^L$ in the propagation phase:
>
>      - Intuitively, considering the optimization targets of the margin loss and TD losses are different, keeping the margin loss with the TD losses in the propagation phase would be beneficial.
>
>      - A deeper explanation for this is: **RL losses are optimized with off-policy trajectories and the newly provided corrective actions from the labeler are never taken in those trajectories**. More concretely:
>         - during the data-collection phase in the $(i+1)$-th iteration, labelers provide action labels $a^L_{t_i}$ on $(s_{t_i},a_{t_i},s_{t_i+1})$ generated from $Q^{\theta_i}$
>         - $\rightarrow$ if removing the margin loss, TD losses optimized on off-policy trajectories (especially those generated from $Q^{\theta_i}$) will not know how to optimize $Q(s_{t_i},a^L_{t_i})$, since $(s_{t_i},a^L_{t_i},s_{t_i}')$ are not available
>        - $\rightarrow$ there is no guarantee that $Q^{\theta_{i+1}}$ would learn from $a^{L}_{t_i}$ after the propagation phase.
>
>   - (2) Effect of the margin loss with $a^{TGT}$ in the propagation phase:
>
>     Recall that (i) $\mathcal{D}^{env}$ contains a large amount of off-policy trajectories generated from previous policies $Q^{\theta_{<i}}$, and (ii) the latest target $\bar{Q}$ is likely to perform better than $Q^{\theta_{<i}}$. So as we explained in Section3 around Line 264-269  "training with pseudo-labels can reduce the cost of collecting human labels by leveraging the large number of unlabeled states in $\mathcal{D}^{env}$. Pseudo-labels can be generated, using predicted greedy actions from the target $\bar{Q}$ on unlabeled states, ..."
>
> - **Experimental ablations for the propagation phase**
>
>   According to the reviewer's concern, there should be three settings to ablate the margin loss in the propagation phase: (1) Removing $a^{TGT}$ but keeping $a^L$ in the propagation phase, (2) Removing $a^{L}$ but keeping $a^{TGT}$ in the propagation phase, and (3) Removing both $a^L$ and $a^{TGT}$ from the propagation phase.
>
>   - For setting (1), which is doubted by the review from comment [1.3], we want to respectfully correct the reviewer that we have already conducted this ablation, denoted as "w/o $a^{TGT}$", in our original paper (Table 2 for *Atari* and Figure 8 for *highway*), and the corresponding experimental analysis is in Section 5.2's last paragraph: ICoPro consistently outperforms, or if not, then performs similarly, with w/o $a^{TGT}$.
>
>     Besides the analysis there, here we give two more concrete summarization about cases that w/o $a^{TGT}$ worse than ICoPro:
>       - (i) when the action space is large (e.g., Atari environments with full action dimensions: Boxing, Battlezone, Frostbite, Alien, Hero)
>       - or (ii) when the number of feedback is not enough (e.g., MsPacman with small budget size)
>
>   - For setting (2) and (3), although these two settings are unreasonable considering the necessity of using $a^L$ as we analyzed above (i.e., Methodological analysis-(1)), we complement the two ablations on *highway* and two Atari environments (*Boxing* and *Seaquest*). We use "w/o Prop's $a^L$" to denote the setting (2), and "w/o Prop's MG" for (3). The experimental results are updated in Appendix C.3.1. The experiments demonstrate that ICoPro's performance significantly decreases in all environments under the two settings, which strongly supports our above analysis.
>
> > [1.4]. IIUC the "w/o aTGT" ablation is the experiment ablating the marginal loss in propagation correct?
>
> No, as we explained above and in our paper's Section 5.1, "w/o $a^{TGT}$" ablation is an ablation that only removes the margin loss with $a^{TGT}$ but keeps the margin loss with $a^L$ in the propagation phase.

---

> ### Author Response · Authors · 2024-11-18
> **About W1-2 & Q1-3 (Part2)**
>
> ## 2. About ablation studies
> > [2.1]. The ablations study are presented in Section 5.2 and Table 2 while in Appendix A, while there is no discussion on the reasons.
>
> We put training plots for Table 2's results in Appendix A due to the space constraint of the main text.
>
> > [2.2]. The author just put the results there. All ablation studies are conducted in the Atari, and further discussion is needed. Why do ablations fail? What insights can we get from the ablation studies?
>
> We want to respectfully correct the reviewer that the ablation studies have not only been conducted in *Atari*, but also in *highway*, in our original paper. Here we think the reason leading to this misunderstanding is the missing two ablations "w/o Finetune" and "w/o $a^{TGT}$" in Figure 2b for highway, which has been explained in our original paper's main text in Section 5.1.'s last paragraph: "In the highway environment, the two ablations perform similarly to ICoPro, and we include them in Appendix A.2 instead of Figure 2b to make it clearer to check."
>
> About the performance of the two ablations ("w/o Finetune" and "w/o $a^{TGT}$") in *highway*:
> - For the "w/o $a^{TGT}$" setting the ablation seems to fail since the performance is almost the same as ICoPro, but as we explained above in "Experimental ablations for the propagation phase", the reason is that highway's action space is small but "w/o $a^{TGT}$" shows its beneficial when the action space is large. Similar ablation fail can also be observed in Atari games with small action space, e.g., *Pong* and *Freeway*.
> - For the "w/o Finetune" setting, the ablation does not fail (i.e., "w/o Finetune" decreases the performance of ICoPro) and can be checked in Figure 8.
>
> ## 3. About human cost / human efficiency
> >[3.1]. In Sec 5.2 the ablation studies only show the performance in tasks but the other metrics such as "human efficiency" are not compared
>
> Our experiments use a fixed number of queries per iteration ($N_q$) and corrective actions per query ($N_{CF}$), and we keep the two hyperparameters the same for all methods, so given any target level of performance, human efficiency (i.e., the number of action labels) can be directly checked in our plots: x-axis refers to the index of training iterations $i$, then $i\times N_q\times N_{CF}$ is the number of action labels cost so far.

---

> ### Author Response · Authors · 2024-11-18
> **About W1-2 & Q1-3 (Part3)**
>
> ## 4. About experiments with non-optimal labelers and the effect of bad actions from the labeler
> > [4.1]. How well does the method work with human labeler in different optimality?
>
> > [4.2]. Q2: How to ensure the label quality? In more complex task like driving, human labeler is hard to understand the outcome of their "corrective actions". For example to rescue a car from crashing left car, the human labeler might steer the car all to the right hand side. And this action though seems to be correct, but it's impossible to be applied in real world (you can't rotate the steering wheel immediately from left to right), and it will also cause unknown outcome such as car rollover. There is no guarantee on the quality of the data especially in the offline setting. In motivation the authors suggest this paper can work with suboptimal human labeler but **no experiment shows the method still work on suboptimal experts.**
>
> 1. We want to respectfully correct the reviewer that the **experiments involving *non-optimal simulated labelers* have been demonstrated in Section 5.3 in our original paper**, and we give extra detailed analysis in Appendix A.1.3. To sum up, ICoPro outperforms non-optimal labelers in most cases.
>
>     Considering that **real human labelers** tend to become better and better at achieving the task or providing corrective actions as they participate in experiments, it's difficult to determine their level of non-optimality concretely. So we do not conduct this set of experiments with real humans. See more discussions in W3 about experiments with real humans.
>
> 2. **Here we give an analysis of *the effect of bad actions from labeler*** and show that **our framework is robust to such actions**:
>
>     Corrective actions provided by a human labeler may be non-optimal in the sense taking $a_t^L$ in $s_t$ leads to an undesired state $s_{t+1}$ (e.g., if taking the reviewer's question as an example, then $a_t^L=TurnRight$ when $s_t=CarIsTurningLeftInWrongPlace$  and then $s_{t+1}=CarRollover$). Considering that giving penalties for undesired states is a common choice, we assume that we could have a (non-optimal) proxy reward that gives a negative reward to the forbidden state $s_{t+1}$, therefore $\widetilde{r}(s_t,a_t^L)=-C$ where $C<0$. Then:
>     - even though the margin loss (eq.1) encourages $a_t^L=\arg\max_{a\in\mathcal{A}} Q^{\theta_i}(s_t,\cdot)$ after the training of the $i-$th iteration
>     - $\rightarrow$ in some following training iteration $i+k$'s data collection phase, we obtain this transition $(s_t, a_t^L, s_{t+1})$ using the greedy policy with respect to $Q^{\theta_{i+k-1}}$
>     - $\rightarrow$ the TD loss (eq.2-3) encourages $Q^{\theta_{i+k}}(s_t,a_t^L)$ to be small since the target $(\widetilde{r}(s_t,a_t^L)+(1-d)\cdot\gamma\max_{a'}\bar{Q}(s_{t+1},a'))=\widetilde{r}(s_t,a_t^L)$ is negative, where $d$ indicates an episode end and is implemented in our code
>     - $\rightarrow$ Q values for state-action pairs in trajectories leading to state $s_t$ will also decrease
>     - $\rightarrow$ **The policy learns to avoid the undesired state $s_t$, so bad effect of $a^L_t$ will not show in $Q^{\theta_{i+k}}$**

---

> ### Author Response · Authors · 2024-11-18
> **About Weakness 3 & 4**
>
> # W3: About experiments with real humans
> > As a paper studying human-in-the-loop policy learning, there is no enough weights on real-human experiments. Section 5.4 shows the experiment with real human subject. I will expecting far more information about this experiment.
>
> - Note that **it is hard to have a fair comparison with baselines with real humans, since we can not guarantee the same performance of human labelers**.
>   - Even though letting the same human labeler conduct experiments for all baseline methods is not fair, because the labeler's performance would become better and better on achieving the task or providing corrective actions as s/he participates in more experiments.
>
> - We believe that **our experiments with *real humans* already demonstrate the feasibility of our novel idea of combining two imperfect signals to train an agent**.
>   - In particular, the primary goal of our experiments with real humans is to demonstrate what kind of performance our framework can achieve by combining corrective actions with proxy rewards, which has been illustrated with the two representative environments (Highway and Pong).
>
> - **Our extensive experiments with *simulated labelers* allow us to better understand and analyze our proposed method in controlled settings**, covering various environments with different proxy rewards, large or small label budgets, and different levels of non-optimality of labelers.
>
> - Having said that, **to further demonstrate our method with human labelers, we select another representative Atari game, *Seaquest*, as a complementary for our user study to further strengthen our user study to show the applicability of ICoPro in complex environments**.
>   - *Seaquest* is an environment with a larger action dimension ($|\mathcal{A}|$=18) than Highway ($|\mathcal{A}|$=5) and Pong ($|\mathcal{A}|$=6), and its proxy reward is non-optimal in the sense that part of the desired performance is not rewarded. Concretely, the game lets the player rescue divers and shoot fish, the player will lose a life if s/he does not rescue drivers within a certain time period, but only shooting fish is rewarded.
>   - ICoPro-human can rescue up to 6 divers (and 3.2 on average) mong 50 evaluation episodes with 400 real human labels and 100K environmental timesteps. In contrast, our simulated Rainbow expert, trained with 46 times more timesteps, can only rescue a maximum of 4 divers (and 1.9 on average).
>
>   We update the experimental details of ICoPro-Human on *Seaquest* into Section 5.4 and Appendix E, as well as a video demonstration in our supplementary materials as described in Appendix E.1.
>
> > Why don't put the evaluation results of the trained policies in each iteration of the training and compare it against the baselines? Is there qualitative results showing the human's corrective actions and the agent's actions?
>
> For the required extra experimental materials, **"the evaluation results of the trained policies in each iteration of the training" (although not with baselines as we explained before) and "qualitative results showing the human's corrective actions and the agent's actions" are updated** into Appendix E.2-E.3 in our newly updated pdf."
>
> # W4: About applicability with real humans
> >Following question 3, providing corrective actions is actually very boring and time-consuming. The setting might be even more exhausting than online human supervision. I doubt whether the method is applicable at all in real-human setting.
>
> - As said in the reviewer's Strengths part, existing human-in-the-loop methods suffer from the cost of human subjects, as they need to be very attentive in an online human-AI shared control setting, while our method adopts the offline setting and utilizes the proxy reward, which is an interesting way to scale up the human-in-the-loop method. Besides the praised offline setting, our method doesn't require the labeler to provide labels for all unsatisfied transitions, which is much less labour-intensive than existing solutions involving action demonstrations.
>
> - Moreover, according to our experiments with real humans, providing corrective actions is user-friendly, especially because they can easily select parts of unsatisfied states to correct instead of all unsatisfied states.

---

> > ### Comment · Reviewer_QvY8 · 2024-11-22
> >
> > Overall, I appreciate the detailed responses and revision to the paper. I've raised my score. I hope the responses and revisions can be shown in the camera ready version. Good luck.

---

> > > ### Author Response · Authors · 2024-11-23
> > >
> > > Thank you, we deeply appreciate the effort and time you've dedicated to providing us with your valuable feedback!

---

### Official Review · Reviewer_CFrc · 2024-11-03

**Soundness:** 3
**Presentation:** 3
**Contribution:** 2
**Rating:** 3
**Confidence:** 4

**Summary:**

This paper presents a new pipeline to learn from corrective actions and proxy rewards, which is interesting and useful for the practical use of RL algorithms. The pipeline of the algorithm is reasonable. The experiments show the effectiveness of the proposed methods.

**Strengths:**

The general idea of the paper is interesting. Involving human labelers in the training process is a very promising way to effectively enhance the training process. The paper is easy to follow.

**Weaknesses:**

One major weakness is that the experiments are kind of weak to support learning from corrective actions. The paper only provides two environments that are involved by real human labeler (Pong and Highway). Most of the environments are using simulated labeler. More environments that show the necessity to involve human labels would be great to support the idea of the paper.



The writing of the paper could be improved.

- The abstract could be further condensed.

- Line 178~185: $\mathcal{S}, \mathcal{A}$ is not defined. The $Q$ function is not formally defined.

- Line 182: Is $\pi^*$ a typo? Do you refer $\pi^*$ to an optimal policy, or greedy policy, or arbirtrary policy?

- Line 353: You should use $a^L \sim $ rather than $a^L=$ if the action is sampled from some $\epsilon$-greedy distribution.



The motivation for combining the two sources of signals is good, but the description in the paper (from Line 66~80) does not make sense to me. The paper stated 'In contrast, our key insight is that the two sources of signals can complement each other.. while the effects of suboptimal corrective actions may be weakened by proxy rewards'.  However, the proxy rewards could also be suboptimal, as it's only an approximation of the real reward function.

**Questions:**

Could you please elaborate more practical examples that involving human labelers is necessary?

**Details Of Ethics Concerns:**

No Ethics Concerns.

---

> ### Author Response · Authors · 2024-11-18
> **Thank you for your feedback, and about Weakness 1 and Question 1**
>
> Thank you for your time, especially for improving the notations in Section 3.  We are deeply grateful for the reviewer's recognition that our general idea is interesting and the training process is promising.
>
> **Here we first recall the review comments with quotes, then give our feedback. For the mentioned updates to the pdf below, we use the blue color in the latest paper pdf to denote updated content.**
>
> We sincerely hope that our response addresses the raised concerns, and we are more than willing to discuss if there are any remaining questions.
>
> # W1 & Q1: About experiments with real humans
> > W1: One major weakness is that the experiments are kind of weak to support learning from corrective actions. The paper only provides two environments that are involved by real human labeler (Pong and Highway). Most of the environments are using simulated labeler. More environments that show the necessity to involve human labels would be great to support the idea of the paper.
>
> > Q1: Could you please elaborate more practical examples that involving human labelers is necessary?
>
> - Note that **it is hard to have a ***fair comparison*** with baselines with real humans**, since we can not guarantee the same performance of human labelers.
>
>   - Even though letting the same human labeler conduct experiments for all baseline methods is not fair, because the labeler's performance would become better and better on achieving the task or providing corrective actions as s/he participates in more experiments.
>   - It is quite common for works involving online human feedback (e.g., [1][2][3][4]) to primarily use simulated experts/labelers for the bulk of the experiments, then validate the performance with real humans in representative environments, like what we have done (we update one additional environment for our user study, see below).
>     - Mentioned References
>
>       [1] PEBBLE: Feedback-Efficient Interactive Reinforcement Learning via Relabeling Experience and Unsupervised Pre-training. ICML 2021
>
>       [2] Efficient Learning of Safe Driving Policy via Human-AI Copilot Optimization. ICLR 2022
>
>       [3] Guarded Policy Optimization with Imperfect Online Demonstrations. ICLR 2023
>
>       [4] RLIF: Interactive Imitation Learning as Reinforcement Learning. ICLR 2024
>
> - We believe that **our experiments with *real humans* already demonstrate the feasibility of our novel idea of combining two imperfect signals to train an agent**.
>   - In particular, the primary goal of our experiments with real humans is to demonstrate what kind of performance our framework can achieve by combining corrective actions with proxy rewards, which has been illustrated with the two representative environments (*Highway* and *Pong*). The two examples on *Highway* and *Pong* have already been practical examples showing that involving human labelers is necessary to achieve the desired performance.
>
> - **Our extensive experiments with *simulated labelers* allow us to better understand and analyze our proposed method in controlled settings**, covering various environments with different proxy rewards, large or small label budgets, and different levels of non-optimality of labelers.
>
> - Having said that, to further demonstrate our method with human labelers, **we selecte another representative Atari game, *Seaquest*, as a complementary for our user study to further strengthen our user study to show the applicability of ICoPro in complex environments.**
>   - *Seaquest* is an environment with a larger action dimension ($|\mathcal{A}|$=18) than *Highway* ($|\mathcal{A}|$=5) and *Pong* ($|\mathcal{A}|$=6), and its proxy reward is non-optimal in the sense that part of the desired performance is not rewarded. Concretely, the game lets the player rescue divers and shoot fish, the player will lose a life if s/he does not rescue drivers within a certain time period, but only shooting fish is rewarded.
>   - ICoPro-human can rescue up to 6 divers (and 3.2 on average) among 50 evaluation episodes with 400 real human labels and 100K environmental timesteps. In contrast, our simulated Rainbow expert, trained with 46 times more timesteps, can only rescue a maximum of 4 divers (and 1.9 on average).
>
>   We update the experimental details of ICoPro-Human on *Seaquest* in Section 5.4 and Appendix E, as well as a video demonstration in our supplementary materials as described in Appendix E.1.

---

> ### Author Response · Authors · 2024-11-18
> **About Weakness 2**
>
> # W2: About writings
> > The writing of the paper could be improved.
> > 1. The abstract could be further condensed.
> > 2. Line 178~185: $\mathcal{S}$, $\mathcal{A}$ is not defined. The $Q$ function is not formally defined.
> > 3. Line 182: Is $\pi^*$ a typo? Do you refer  $\pi^*$ to an optimal policy, or greedy policy, or arbirtrary policy?
> > 4. Line 353: You should use $a^L\sim$ rather than $a^L=$ if the action is sampled from some $\epsilon$-greedy distribution.
>
> Thank you for these comments. We have fixed points 2-4 (see blue correction in the updated manuscript).
>
> Regarding the abstract, we have carefully condensed our abstract before submission to ensure it fully meets the requirements of ICLR. But if the reviewer has any concrete suggestions for improving the abstract, please do not hesitate to let us know and we are more than willing to make further improvements.

---

> ### Author Response · Authors · 2024-11-18
> **About Weakness 3**
>
> # W3: About explanations of learning from non-optimal corrective actions and proxy rewards
> > The motivation for combining the two sources of signals is good, but the description in the paper (from Line 66~80) does not make sense to me. The paper stated **'In contrast, our key insight is that the two sources of signals can complement each other. ... while the effects of suboptimal corrective actions may be weakened by proxy rewards'**. However, the proxy rewards could also be suboptimal, as it's only an approximation of the real reward function.
>
> We think the reviewer's concern raised here comes from the neglect of a key subsentence (highlighted with ***bold italic*** fonts below) in the quoted sentences. In lines 74~75, we stated "the two sources of signals can complement each other. ***Since they are generally imperfect in different state-space regions***, ...", in other words, **the two learning signals usually perform well / poorly in different sets of states**.
>
> To further clarify this point, we provide **two concrete examples**, among which **the example (ii) address the reviewer's question about *why the effects of suboptimal corrective actions may be weakened by proxy rewards***:
>
>   - **(i).** If (non-optimal) corrective actions **have effects on the different set of states** with the (non-optimal) proxy rewards:
>
>     - Many goal-conditioned rewards used in practice are non-optimal proxy rewards, which reward reaching the expected goal states without characterizing the desired way to reach goals into the reward (due to the difficulty of designing such rewards). Then **there could be many solutions to maximize such a proxy reward, and corrective actions help to express the desired performance into the policy while maximizing the proxy reward**. Those corrective actions could come from non-optimal labelers since they may not manage to reach the goal themselves (e.g., catch the ball), but their actions guide the agent to achieve it (e.g., guide the agent to move closer to the ball when it's coming).
>
> - **(ii).**  If (non-optimal) corrective actions **have effects on the same set of states** with the (non-optimal) proxy rewards:
>
>   **In this case, it's fine if the corrective action is optimal, so we mainly analyse the performance of our framework with respect to a non-optimal corrective action.** For example,
>   - Corrective actions provided by a human labeler may be non-optimal in the sense taking $a_t^L$ in $s_t$ leads to an undesired state $s_{t+1}$ (e.g., $a_t^L=SpeedUpForward$ when $s_t=CloseToTheFrontCar$ and then $s_{t+1}=CarCrash$). Considering that giving penalties for undesired states is a common choice, we assume that we could have a (non-optimal) proxy reward that gives a negative reward to the forbidden state $s_{t+1}$, therefore $\widetilde{r}(s_t,a_t^L)=-C$ where $C<0$. Then:
>     - even though the margin loss (eq.1) encourages $a_t^L=\arg\max_{a\in\mathcal{A}} Q^{\theta_i}(s_t,\cdot)$ after the training of the $i-$th iteration
>     - $\rightarrow$ in some following training iteration $i+k$'s data collection phase, we obtain this transition $(s_t, a_t^L, s_{t+1})$ using the greedy policy with respect to $Q^{\theta_{i+k-1}}$
>     - $\rightarrow$ the TD loss (eq.2-3) encourages $Q^{\theta_{i+k}}(s_t,a_t^L)$ to be small since the target $(\widetilde{r}(s_t,a_t^L)+(1-d)\cdot\gamma\max_{a'}\bar{Q}(s_{t+1},a'))=\widetilde{r}(s_t,a_t^L)$ is negative, where $d$ indicates an episode end and is implemented in our code
>     - $\rightarrow$ Q values for state-action pairs in trajectories leading to state $s_t$ will also decrease
>     - $\rightarrow$ **The policy learns to avoid the undesired state $s_t$ and the bad effect of $a^L_t$ will not show in $Q^{\theta_{i+k}}$**

---

> ### Author Response · Authors · 2024-11-22
> **Friendly reminder**
>
> Dear reviewer CFrc,
>
> We deeply appreciate the effort and time you've dedicated to providing us with your valuable feedback. We believe we have addressed all your concerns. If you have any other concerns, please do not hesitate to let us know and we will be happy to address them.
>
> Best, Authors

---

> ### Author Response · Authors · 2024-11-25
> **Complementary response for Question1**
>
> Considering that our previous response to Reviewer CFrc's Question 1 may not have fully addressed the concern raised, we hope to take this opportunity to provide a complementary response to the original message.
>
> > Q1: Could you please elaborate more practical examples that involving human labelers is necessary?
>
> Our initial user studies on *Pong* and *Highway* serve as two representative scenarios to demonstrate that **involving human labelers is necessary for obtaining an agent that finishes a task with the desired performance**. **This requirement is common and reflects numerous practical applications beyond the two specific environments we present**. Human labelers play an important role in achieving this requirement in the two scenarios:
> - **(i) When the desired performance style cannot be expressed by the reward function (e.g., in *Pong*, achieving human-like behavior is a challenge that goes beyond what a simple reward function can convey)**, involving human labelers is necessary to express their expected performance.
>   - Although imitation learning from demonstrations may achieve similar outcomes, there are common instances that involving human action labelers is indispensable, as we said in Line 63-65 in our manuscript: "Regarding corrective actions, in contrast to typical demonstrations of whole trajectories, this feedback is usually much easier for the labeler to provide, since **humans may not be able to complete a task themselves but can readily offer action preference on some states.**"
> - **(ii) When the desired performance style may be engineered into the reward function in some way (e.g., in *Highway*, the desired performance style may be partially integrated into the proxy rewards)**, involving human labelers is necessary to improve the learning efficiency and obtain the desired performance stably.
>   - As we said in Line 69-72, "**solely learning from proxy rewards would either lead to very slow learning** (e.g., when proxy rewards are well-aligned with ground-truth rewards but are very sparse) **or yield a policy whose performance is not acceptable to the system designer (e.g., when proxy rewards are dense, but misspecified)**". Our experiments have illustrated such performance with the Rainbow baseline (e.g., Figure 2a, Table 2).
>
> Our experiments on *Pong* and *Highway* show the benefits of involving human labelers for the above-mentioned representative scenarios, and *Seaquest* further confirms the applicability of our method in more complex environments.

---

> ### Comment · Area_Chair_61jF · 2024-11-25
> **Please read rebuttal**
>
> Dear Reviewer CFrc, Could you please read the authors' rebuttal and give them feedback at your earliest convenience? Thanks. AC

---

> ### Author Response · Authors · 2024-11-26
> **A friendly reminder**
>
> Dear reviewer CFrc,
>
> We highly appreciate your time and the great efforts you have put into reviewing our work. We have revised some of the previous rebuttal content to better address your concerns and questions.
>
> If possible, we hope to further engage in detailed technical communication to earn your endorsement. We are eagerly looking forward to hearing whether you have any additional concerns or suggestions for our work.
>
> Best, Authors

---

### Official Review · Reviewer_BJFG · 2024-11-03

**Soundness:** 3
**Presentation:** 3
**Contribution:** 2
**Rating:** 5
**Confidence:** 3

**Summary:**

In this paper, a framework is proposed where a human labeler can offer additional feedback in the form of corrective actions, reflecting the labeler's action preferences, even though this feedback may also be imperfect. Within this context, a novel value-based deep RL algorithm, Iterative Learning from Corrective actions and Proxy rewards (ICoPro), is introduced. This algorithm follows a three-phase cycle: (1) Eliciting sparse corrective actions from a human labeler on the agent's demonstrated trajectories; (2) Integrating these corrective actions into the Q-function using a margin loss to ensure adherence to the labeler's preferences; (3) Training the agent with standard RL losses, augmented with a margin loss to learn from proxy rewards and propagate the Q-values acquired from human feedback.

**Strengths:**

This paper explores an intriguing imitation learning problem in which human demonstrations are imperfect, although some corrective actions may be available. Additionally, there is no explicit reward signal, but certain action preferences (raw reward) are provided.

**Weaknesses:**

1) The performance of the human feedback policy would significantly impact the total performance of the proposed method. For example, if the human feedback is from an expert policy, then the cloning from its behaviors is reasonable and helpful for improving the performance. But if the human feedback is from a sub-optimal or even a random policy, then its behaviors may be harmful. Therefore, what we concern is how the level of sub-optimality of the human feedback policy would impact the performance of the algorithm.
	Currently, in Section 5.3 the authors have simulated the sub-optimal human feedback by inserting noisy actions, but we think such simulation is not enough for the explanation for this problem. Maybe the authors could utilize a half-trained Rainbow policy to generate the imperfect behaviors.

	2) The positions (states) of human feedbacks are also important for the efficiency and effectiveness. We observe that the authors selected the feedback positions with the method in RLIF [1], but we does not see the comparison and discussion with RLIF.
	Besides, to be fair, maybe this trick should also be utilized to other methods for comparison introduced in this paper.

[1] RLIF: Interactive imitation learning as reinforcement learning.

	3) The working scenarios. In our opinion, this method could be seen as a method that online learning while cloning from a specific behavior policy (human feedback). So why such combination could solve the problem of sub-optimality of the proxy reward and the human feedback? For example, if the proxy reward and human feedback make the similar mistake, then such combination would not work. Clearly, in such scenarios, the proposed method could achieve the claimed performance.

	4) In this paper, the authors raised an interesting and inspiring problem. However, the problem lacks of mathematical formulation and theoretical evidence for its effectiveness.

       5)  I found that the methodology adopted in this paper is closely related to those references cited in line 229 and the difference to each of those should be clearly stated. In addition, if the range of reward is not limited, then the loss of  eq.1 could be negative, as this expression is different from that of the traditional margin loss.

**Questions:**

see above

---

> ### Author Response · Authors · 2024-11-18
> **Thank you for your feedback, and about Weakness 1**
>
> Thank you for your constructive insights and meticulous feedback!
>
> **Here we first recall the review comments with quotes, then give our feedback. For the mentioned updates to pdf below, we use the blue color in the latest paper pdf to denote updated content.**
>
> We sincerely hope that our response clearly addresses the raised concerns, and we are more than willing to discuss any remaining questions.
>
> # W1: About the experiments with labelers at different levels of performance
> > The performance of the human feedback policy would significantly impact the total performance of the proposed method. For example, if the human feedback is from an expert policy, then the cloning from its behaviors is reasonable and helpful for improving the performance. But if the human feedback is from a sub-optimal or even a random policy, then its behaviors may be harmful. Therefore, what we concern is how the level of sub-optimality of the human feedback policy would impact the performance of the algorithm. Currently, in Section 5.3 the authors have simulated the sub-optimal human feedback by inserting noisy actions, but we think such simulation is not enough for the explanation for this problem. Maybe the authors could utilize a half-trained Rainbow policy to generate the imperfect behaviors.
>
> We **have already executed a set of experiments in our original paper, named *DiffLabeler*, that utilized some half-trained Rainbow policies to generate behaviors with different levels of imperfection**, as explained in Section 5.3 (Figure 3c), and Appendix A.1.3. Besides the previous existing experiments on *highway* and two Atari games (*Boxing* and *Seaquest*),  we further update Appendix A.1.3 with one extra Atari game (*Pong*) to give a more convincing analysis of this setting. **Our experiments of *DiffLabeler* demonstrate that**:
>
> (i). For ICoPro, a worse labeler does not necessarily decrease its performance.
>   - For example, as shown in Figure 7b, the two labelers with performance ranks 0 and 1 (in green and nacarat) in Boxing-$N_q=5$, or the two labelers with performance ranks 2 and 3 (in purple and pink) in Seaquest-$N_q=20$, ICoPro labeled by them exhibit similar performances, despite the obvious performance gap between these two labelers.
>
> (ii). Labelers with lower performance indeed tend to lead to worse performances of the trained agent from ICoPro, which is also widely noted by related works that involve human feedback (e.g., PVP[6], RLIF[7]).
>
> (iii). ICoPro demonstrates a strong capability to overcome the non-optimality of the labeler in most cases:
>   - For example, in *Pong*, even though the worst labeler only achieves scores around -15, ICoPro trained with this labeler can still achieve scores larger than +10.
>   - But if the labeler's performance is bad, a larger number of "corrective" actions (i.e., experiments using larger $N_q$) may lead to a poorer performance of ICoPro as shown in Figure 7b.

---

> ### Author Response · Authors · 2024-11-18
> **Abou Weakness 2**
>
> # W2: Comparison with RLIF
> > The positions (states) of human feedbacks are also important for the efficiency and effectiveness. We observe that the authors selected the feedback positions with the method in RLIF [1], but we does not see the comparison and discussion with RLIF. Besides, to be fair, maybe this trick should also be utilized to other methods for comparison introduced in this paper.
> [1] RLIF: Interactive imitation learning as reinforcement learning.
>
> This comment raised by the reviewer can be separated into two parts, and we address them one by one:
>
> 1. We **totally agree with the reviewer that this trick should be utilized in other methods for comparison introduced in this paper, and that's what we have done when comparing with baselines** as mentioned in Section 5.1's Baselines paragraph. More concretely, we use the same criterion for all baselines, as explained in Section 5.1's Simulated Lebeler paragraph, to decide which states to correct.
>
> 2. About the comparison with **RLIF**:
>   - In our original manuscript, we did not include it as one of the baselines because their learning framework is not as suitable as for incorporating both corrective actions and proxy rewards as the other baselines.
>   - But we are willing to add this part of the experiments to compare our method (ICoPro) with RLIF: Here **we first recall the design of RLIF and compare it with ICoPro and other baselines (see (i) below), then we explain our experimental setting for RLIF and analyse the results (see (ii) below)**.
>
>     **(i). Discussions about RLIF**:
>
>     - Both RLIF and ICoPro can be seen as iterative learning methods: Inside each iteration, we first collect human feedback on trajectories generated from the current policy, then update the policy with the updated dataset.
>
>     - For **feedback type**, RLIF is the same as PVP which lets the labelers **online** oversee the training progress and **take over control** from unsatisfied states to satisfied states. As we explained in the original manuscript, such interaction can be labor-intensive while ICoPro does not require labelers to correct all bad state-action pairs.
>
>     - For **policy learning**, RLIF directly uses a -1 reward for corrected states and 0 otherwise and then lets the RL algorithm learn from such replay buffer, while ICoPro and PVP [6] involve regularizing the Q-values directly with action feedbacks.
>
>     - For **simulated experts / labelers**, both ICoPro and RLIF use pre-trained checkpoints' $Q$-values' to guide the intervention but for different intentions:
>       - RLIF uses $Q(s,\pi^{labeler}(s))>Q(s,\pi(s))+\delta$ as a condition to determine if an online state should be intervened, so "intervene in the current $s-a$ pair as long as the labeler is confident enough";
>       - ICoPro selects states with top Q-differences from an offline segment with $Q(s,a^{labeler})-Q(s,a)$, so "provide corrective actions for the $s-a$ pairs in this segment that the labeler is most unsatisfied with".
>
>     **(ii). Experimental comparisons with RLIF**:
>
>     - To have a fair comparison of RLIF with ICoPro and other baselines, we implement their core design, which (1) directly uses a -1 reward for corrected states and 0 otherwise, and (2) lets the RL algorithm learn from such replay buffer, into the same learning framework as ICoPro and other baselines.
>
>     - We put more details about the experimental setups and the results on *highway* and two Atari games (*Boxing* and *Seaquest*) of RLIF into Appendix C.2.1.
>         - In all the three environments, **RLIF's learning curves are almost flat, indicating its incompetent in learning meaningful policies with offline corrective actions, as shown in Figure 11 in Appendix C.2.1**.
>
>         Such results strongly support the necessity of regularizing the Q-values directly from the action labels, which is the design adapted in our method and our two SOTA baselines (DQfD [4] and PVP [6]), instead of simply putting the reward for that action into the replay buffer as RLIF [7].

---

> ### Author Response · Authors · 2024-11-18
> **About Weakness3**
>
> # W3: About the mechanism and working scenarios combining non-optimal corrective actions and proxy rewards
> > The working scenarios. In our opinion, this method could be seen as a method that online learning while cloning from a specific behavior policy (human feedback). So why such combination could solve the problem of sub-optimality of the proxy reward and the human feedback? For example, if the proxy reward and human feedback make the similar mistake, then such combination would not work. Clearly, in such scenarios, the proposed method could (typo? could not?) achieve the claimed performance.
>
> - First, we agree with the reviewer that *"if the proxy reward and human feedback make the similar mistake, then such combination would not work"*. However, this scenario is not the focus of the paper, as it is highly unlikely in realistic settings for two signals of different natures to be *always* incorrect in the same states.
>
> - Second, we have mentioned the intuitions for *"why such combination could solve the problem of sub-optimality of the proxy reward and the human feedback"* in lines 73-78 in our original manuscript, where we stated
>     >"In contrast, our key insight is that the two sources of signals can complement each other. Since they are generally imperfect in different state-space regions, bad decisions learned from proxy rewards can be corrected by the human labeler, while the effects of suboptimal corrective actions may be weakened by proxy rewards.".
>
>     In other words, our key insight is that the **two sources of signals can complement each other, from the observation that the two learning signals usually perform well / badly in different sets of states**. Here **we further analyse the working mechanism of ICoPro under two representative cases with corresponding examples of this observation**:
>
>   - **(i).** If (non-optimal) corrective actions **have effects on the different set of states** with the (non-optimal) proxy rewards:
>
>     Many goal-conditioned rewards used in practice are non-optimal proxy rewards. These rewards focus on reaching the expected goal states but are unable to adequately characterize the desired procedure for achieving those goals, due to the difficulty of engineering such rewards. Then **there could be many solutions to maximize such a proxy reward, and corrective actions help to express the desired performance into the policy while maximizing these non-optimal proxy rewards**. Those corrective actions could come from non-optimal labelers since they may not manage to reach the goal themselves (e.g., catch the ball), but their actions guide the agent to achieve it (e.g., guide the agent to move closer to the ball when it's coming).
>
>   - **(ii).** If (non-optimal) corrective actions **have effects on the same set of states** (e.g., avoiding a car crash) as the (non-optimal) proxy rewards :
>
>      **In this case, it's fine if the corrective action is optimal, so we mainly analyse the performance of our framework with respect to a non-optimal corrective action.** For example,
>
>     Corrective actions provided by a human labeler may be non-optimal in the sense taking $a_t^L$ in $s_t$ leads to an undesired state $s_{t+1}$ (e.g., $a_t^L=SpeedUpForward$ when $s_t=CloseToTheFrontCar$ and then $s_{t+1}=CarCrash$). Considering that giving penalties for undesired states is a common choice, we assume that we could have a (non-optimal) proxy reward that gives a negative reward to the forbidden state $s_{t+1}$, therefore $\widetilde{r}(s_t,a_t^L)=-C$ where $C<0$. Then:
>     - even though the margin loss (eq.1) encourages $a_t^L=\arg\max_{a\in\mathcal{A}} Q^{\theta_i}(s_t,\cdot)$ after the training of the $i-$th iteration
>     - $\rightarrow$ in some following training iteration $i+k$'s data collection phase, we obtain this transition $(s_t, a_t^L, s_{t+1})$ using the greedy policy with respect to $Q^{\theta_{i+k-1}}$
>     - $\rightarrow$ the TD loss (eq.2-3) encourages $Q^{\theta_{i+k}}(s_t,a_t^L)$ to be small since the target $(\widetilde{r}(s_t,a_t^L)+(1-d)\cdot\gamma\max_{a'}\bar{Q}(s_{t+1},a'))=\widetilde{r}(s_t,a_t^L)$ is negative, where $d$ indicates an episode end and is implemented in our code
>     - $\rightarrow$ Q values for state-action pairs in trajectories leading to state $s_t$ will also decrease
>     - $\rightarrow$ **The policy learns to avoid the undesired state $s_t$ and the bad effect of $a^L_t$ will not show in $Q^{\theta_{i+k}}$.**

---

> ### Author Response · Authors · 2024-11-18
> **About Weakness4:**
>
> # W4: About mathematical formulation and theoretical evidence for its effectiveness
> > In this paper, the authors raised an interesting and inspiring problem. However, the problem lacks of mathematical formulation and theoretical evidence for its effectiveness.
>
> - For **mathematical formulation**:
>
>   - (i) We have introduced the mathematical format of corrective actions in Section 3's second paragraph. To recap: given a query $q=(s_t,a_t, \dots,s_{t+T-1},a_{t+T-1})$, a labeler with a policy $\pi^L$ provides a corrective action $a^L_{t'}=\pi^L(s_{t'})$ on the selected state(s) $s_{t'}$.
>
>   - (ii) For proxy rewards, we do use strict formulation for a broader compatibility with related works. In our extensive experiments, the formats of proxy rewards indeed cover a wide range of common reward types (e.g., dense or sparse, immediate or delayed, etc.).
>
> - For **theoretical evidence** of our framework:
>
>     - (i) First, the backbone framework involves (1) online imitation learning with BC, which is theoretically supported by the DAgger [1] paper to guarantee that the final performance will not be much worse than the labeler, (2) using  RL and IL losses to learn a Q function from two learning signals as we explained in W3, which can be regarded as a regularized deep Q-learning that trace back to Kim 2013 [2] and Piot 2014 [3] that transfer the IL constraints into the RL optimization target with slack variables.
>     - (ii) Second, our response in W3 for this reviewer can also serve as a readily comprehensible theoretical discussion.
>
>
>
> Here we hope to **recall the main intention of this paper is to demonstrate a novel practical learning framework that efficiently learns from both corrective actions and proxy rewards**. Our **extensive empirical results have clearly verified the efficiency of ICoPro** in this framework compared with strong SOTA baseline methods (e.g., DQfD (Hester et al., 2018 [4]) and PVP (Peng et al., 2023 [6])), which also mainly resort to empirical analysis.
>
> Despite these extensive empirical analyses and theoretical efforts, we also agree with the reviewer that conducting more advanced theoretical analysis is an interesting future direction in this domain, as we have mentioned at the end of Section 6 (Conclusions and Limitations).

---

> ### Author Response · Authors · 2024-11-18
> **About Weakness 5**
>
> # W5: About the margin loss used in eq.1 and its related works
> > I found that the methodology adopted in this paper is closely related to those references cited in line 229 and the difference to each of those should be clearly stated. In addition, if the range of reward is not limited, then the loss of eq.1 could be negative, as this expression is different from that of the traditional margin loss.
>
> This comment raised by the reviewer can be separated into two parts, and we address them one by one:
>
> - 1. > this paper is closely related to those references cited in line 229 and the difference to each of those should be clearly stated.
>
>     In that line we cited Kim et al., 2013 [2], Piot et al., 2014 [3], Hester et al., 2018 [4], and Ibarz et al., 2018 [5], the relation to ICoPro is:
>     - For the **human feedback**: The four papers use expert demonstrations to optimize margin loss, while ICoPro uses corrective actions. Note that demonstrations refer to whole trajectories, which are quite different from our corrective actions.
>     - For the **margin loss**: DQfD (Hester et al., 2018 [4]) is a work that uses the same margin loss as Kim et al., 2013 [2] and Piot et al., 2014 [3], but parametrises the Q function with deep neural networks instead of kernel methods. Then Ibarz et al., 2018 [5] is a successor work of DQfD that further replaces the ground-truth reward used in DQfD with learned ones from pairwise comparisons. ICoPro's margin loss is the same as Hester et al., 2018 [4] and Ibarz et al., 2018 [5].
>     - For the **policy learning method**: Compared to ICoPro, the other four methods (1) do not involve two separate learning phases, and they directly add the margin loss with expert demonstrations to the original RL losses; (2) do not use target pseudo labels.
>
> - 2. > if the range of reward is not limited, then the loss of eq.1 could be negative
>
>     We want to respectfully correct the reviewer that the loss of eq.1 can not be negative since the minimal value is 0 when $\arg\max_{a\in\mathcal{A}}Q_{\theta}(s,\cdot)=a^L$, where $\mathcal{A}$ is the discrete action space and $Q_{\theta}:\mathcal{S}\rightarrow \mathbb{R}^{|\mathcal{A}|}$ is the training policy. This minimal value is unrelated to the limit of the reward.

---

> ### Author Response · Authors · 2024-11-18
> **Mentioned References**
>
> # Mentioned References:
>
> [1] Stephane Ross, Geoffrey Gordon, and Drew Bagnell. A reduction of imitation learning and structured prediction to no-regret online learning. In Geoffrey Gordon, David Dunson, and MiroslavDud´ık (eds.), Proceedings of the Fourteenth International Conference on Artificial Intelligence and Statistics, volume 15 of Proceedings of Machine Learning Research, pp. 627–635, Fort Lauderdale, FL, USA, 11–13 Apr 2011. PMLR.
>
> [2] Beomjoon Kim, Amir-massoud Farahmand, Joelle Pineau, and Doina Precup. Learning from limited demonstrations. In C.J. Burges, L. Bottou, M. Welling, Z. Ghahramani, and K.Q. Weinberger (eds.), Advances in Neural Information Processing Systems, volume 26. Curran Associates, Inc., 2013.
>
> [3] Bilal Piot, Matthieu Geist, and Olivier Pietquin. Boosted bellman residual minimization handling expert demonstrations. In Machine Learning and Knowledge Discovery in Databases: European Conference, ECML PKDD 2014, Nancy, France, September 15-19, 2014. Proceedings, Part II 14, pp. 549–564. Springer, 2014.
>
> [4] Todd Hester, Matej Vecerik, Olivier Pietquin, Marc Lanctot, Tom Schaul, Bilal Piot, Dan Horgan, John Quan, Andrew Sendonaris, Ian Osband, et al. Deep q-learning from demonstrations. In Proceedings of the AAAI Conference on Artificial Intelligence, volume 32, 2018.
>
> [5] Borja Ibarz, Jan Leike, Tobias Pohlen, Geoffrey Irving, Shane Legg, and Dario Amodei. Reward learning from human preferences and demonstrations in atari. Advances in neural information processing systems, 31, 2018.
>
> [6] Zhenghao Mark Peng, Wenjie Mo, Chenda Duan, Quanyi Li, and Bolei Zhou. Learning from active human involvement through proxy value propagation. Advances in neural information processing systems, 36, 2023.
>
> [7] Jianlan Luo, Perry Dong, Yuexiang Zhai, Yi Ma, and Sergey Levine. RLIF: Interactive imitation learning as reinforcement learning. In The Twelfth International Conference on Learning Representations, 2024.

---

> ### Author Response · Authors · 2024-11-22
> **Friendly reminder**
>
> Dear reviewer BJFG,
>
> We deeply appreciate the effort and time you've dedicated to providing us with your valuable feedback. We believe we have addressed all your concerns. If you have any other concerns, please do not hesitate to let us know and we will be happy to address them.
>
> Best, Authors

---

> ### Comment · Area_Chair_61jF · 2024-11-25
> **Please read rebuttal**
>
> Dear Reviewer BJFG, Could you please read the authors' rebuttal and give them feedback at your earliest convenience? Thanks. AC

---

> ### Author Response · Authors · 2024-12-01
> **Friendly Reminder: Rebuttal Deadline Approaching**
>
> Dear reviewer BJFG,
>
> We highly appreciate your time and the great efforts you have put into reviewing our work. We have revised some of the previous rebuttal content to better address your concerns and questions.
>
> Considering that we have only about 48 hours left until the rebuttal period ends, we are eagerly looking forward to hearing whether you have any additional concerns or suggestions for our work. If possible, we hope to further engage in detailed technical communication to earn your endorsement.
>
> Best, Authors

---

### Author Response · Authors · 2024-11-24
**Summary of changes**

Dear Reviewers,

Thank you for your valuable comments, which have significantly contributed to the improvement of our paper. We have incorporated your suggestions into the newly uploaded manuscript, with the modifications highlighted in blue. **While we have mentioned these related modifications individually in our responses to each reviewer, we provide a summary below for your convenience.**

Besides some minor revisions enhancing our writing's clarity and readability, the key updates include:
- Add a set of ablation studies for the margin loss with $a^L$ used in the propagation-phase of our method ICoPro (Appendix C.3.1)
- Add the results of comparing ICoPro with RLIF (Appendix C.2.1)
- Add more materials to support our user study, including (1) another complex environment (Section 5.4), and (2) detailed experimental results including training plots, qualitative examples about human feedback (Appendix E)
- Add more experimental examples and analysis for the *DiffLabeler* and *DiffRand* settings, which are two settings with non-optimal labelers (Appendix A.1.3)

Please let us know if you have any other concerns or need further clarification regarding the modifications mentioned above.

Best, Authors

---

### Meta-Review · Area_Chair_61jF · 2024-12-18

**Metareview:**

This paper proposes that human labelers can provide corrective actions and proxy rewards. This is interesting and useful for improving RL algorithms. The algorithm's pipeline is reasonable. The experiments are effective. This algorithm has three phases: (1) human labeling of demonstration trajectories; (2) nudging the corrective action into value estimation; (3) RL with the augmented Q value. The overall framework is novel. The quality is above the bar of ICLR. I hope the authors could improve the presentation and formulation in the camera-ready version.

**Additional Comments On Reviewer Discussion:**

The authors addressed most of the reviewers' comments.

---

### Decision · Program_Chairs · 2025-01-22

Accept (Poster)